# Hypothesis generation for rare and undiagnosed diseases through clustering and classifying time-versioned biological ontologies

**Michael S. Bradshaw**[1], **Connor Gibbs**[2], **Skylar Martin**[1], **Taylor Firman**[3], **Alisa Gaskell**[3], **Bailey Fosdick**[4], **Ryan Layer**[1]\*

**1** Department of Computer Science, University of Colorado Boulder, Boulder, CO, United States of America, **2** Department of Statistics, Colorado State University, Fort Collins, CO, United States of America, **3** Precision Medicine Institute, Children's Hospital Colorado, Aurora, CO, United States of America, **4** Department of Biostatistics & Informatics, Colorado School of Public Health, Aurora, CO, United States of America

\* ryan.layer@colorado.edu

**Data Availability Statement:** Data relating to patients at Children's Hospital Colorado (CHCO) is not publicly available per agreements with CHCO

## Abstract

Rare diseases affect 1-in-10 people in the United States and despite increased genetic testing, up to half never receive a diagnosis. Even when using advanced genome sequencing platforms to discover variants, if there is no connection between the variants found in the patient's genome and their phenotypes in the literature, then the patient will remain undiagnosed. When a direct variant-phenotype connection is not known, putting a patient's information in the larger context of phenotype relationships and protein-protein interactions may provide an opportunity to find an indirect explanation. Databases such as STRING contain millions of protein-protein interactions, and the Human Phenotype Ontology (HPO) contains the relations of thousands of phenotypes. By integrating these networks and clustering the entities within, we can potentially discover latent gene-to-phenotype connections. The historical records for STRING and HPO provide a unique opportunity to create a network time series for evaluating the cluster significance. Most excitingly, working with Children's Hospital Colorado, we have provided promising hypotheses about latent gene-to-phenotype connections for 38 patients. We also provide potential answers for 14 patients listed on MyGene2. Clusters our tool finds significant harbor 2.35 to 8.72 times as many gene-to-phenotype edges inferred from known drug interactions than clusters found to be insignificant. Our tool, BOCC, is available as a web app and command line tool.

## Introduction

Paradoxically, rare diseases are not rare. 25 to 30-million people in the United States are affected by a rare disease that spans the frequency spectrum from sickle cell anemia, which affects 100,000 people in the US, to multicentric carpo-tarsal osteolysis, which affects 60 people

and the IRB. All other data used in this study is available at https://github.com/MSBradshaw/BOCC.

**Funding:** This work was supported by a grant from Children's Hospital Colorado. Members of the funding body collected the patient data, aided in the direction of analysis, and are authors of the paper.

**Competing interests:** The authors have declared that no competing interests exist.

worldwide, all the way down to n-of-1 diseases where a patient's presenting phenotypes and genotypes are the only known case world-wide [1].

Living with a rare disease is extremely challenging for patients and their families. Even getting a diagnosis takes, on average, four to eight years—if they get one at all [2–5]. To receive a diagnosis, at least a partial understanding of the disease mechanism is required, but because these conditions are rare, they are often precluded from getting a diagnosis because of these criteria. In this diagnostic odyssey, patients often endure extensive testing in hopes of finding a connection between their observed genetic variants and apparent phenotypes. Given the difficulty of studying rare genetic diseases, the literature is often limited, and in many cases, there are no known connections. Online platforms such as Matchmaker Exchange [6] and MyGene2 [1] have made great strides in making rare diseases easier to study by connecting rare disease families and growing cohorts, but many cases remain unsolved.

For the families still waiting for a diagnosis, we propose expanding the scope of a patient's medical history to uncover latent interactions that could help guide a diagnosis. Using the STRING protein-protein-interaction (PPI) network [7], we can extend a patient's variant-harboring genes to include all other genes with known protein interactions. Using HPO, we can expand the patient's assigned phenotypes to include all closely related phenotypes and diseases. By connecting STRING and HPO [8] with the gene-to-phenotype (g2p) connections from Orphanet [9] and OMIM [10], we can then look for indirect associations between the patient's assigned data points. For example, MyGene2 patient 1930 has variants of uncertain significance (VUS) in the gene *NBEA* and pseudo gene *SSPO*. (Fig 1A, dark red circle) and presented with Seizures, Myoclonus, Gait ataxia, and 14 other phenotypes (Fig 1A dark blue). Due to a lack of understanding of their VUSes, compounded with no known connection between their phenotypes and affected genes, patient 1930 remains undiagnosed. The next steps for this case would involve experimentally exploring the relationships between the patient's VUS and presenting phenotypes. The amount of time and money required to explore all possible combinations between these two genes and the 17 presenting phenotypes make this task likely unfeasible, necessitating a different approach.

In the expanded gene (Fig 1A, light red circle) and phenotype (Fig 1B, light blue square) sets, there are multiple connections between *NBEA* and Seizures. Each of these connections represents a potential path toward diagnosis.

While STRING, HPO, Orphanet, and OMIM bring power to these analyses, they also add complexity. In particular, integrating these networks creates a large graph with approximately 35,000 nodes (genes and phenotypes) and approximately 6,000,000 edges (a connection between two nodes). In such a densely connected network, there are relatively short paths between all genes and all phenotypes. For example, while only 25% of genes have a known disease association, in the integrated graph 50% of genes are only one hop from an HPO term. In the case of patient 1930, their VUS harboring gene, *NBEA*, is connected to all of their phenotypes by paths of length 1 for a majority of cases and never more than a path of length 2. With the scale and connectivity of this network, simply having a path between two entities is not necessarily meaningful. But if we can identify regions with the graph where nodes are highly related and interconnected, like that shown in Fig 1A, between *NBEA*'s neighbors and Seizures, the connections between them can be meaningful. We show that we can find regions like this in the graph using network clustering algorithms and accurately predict which clusters are most likely to harbor latent gene-to-phenotype (g2p) connections. Genes and phenotypes that co-occur in these clusters are more likely to be connected in the near future. This is not the same as a diagnosis or solving a case but it can greatly reduce the number of gene and phenotype combinations that need to be experimentally confirmed, expediting the diagnostic odyssey of people like patient 1930.

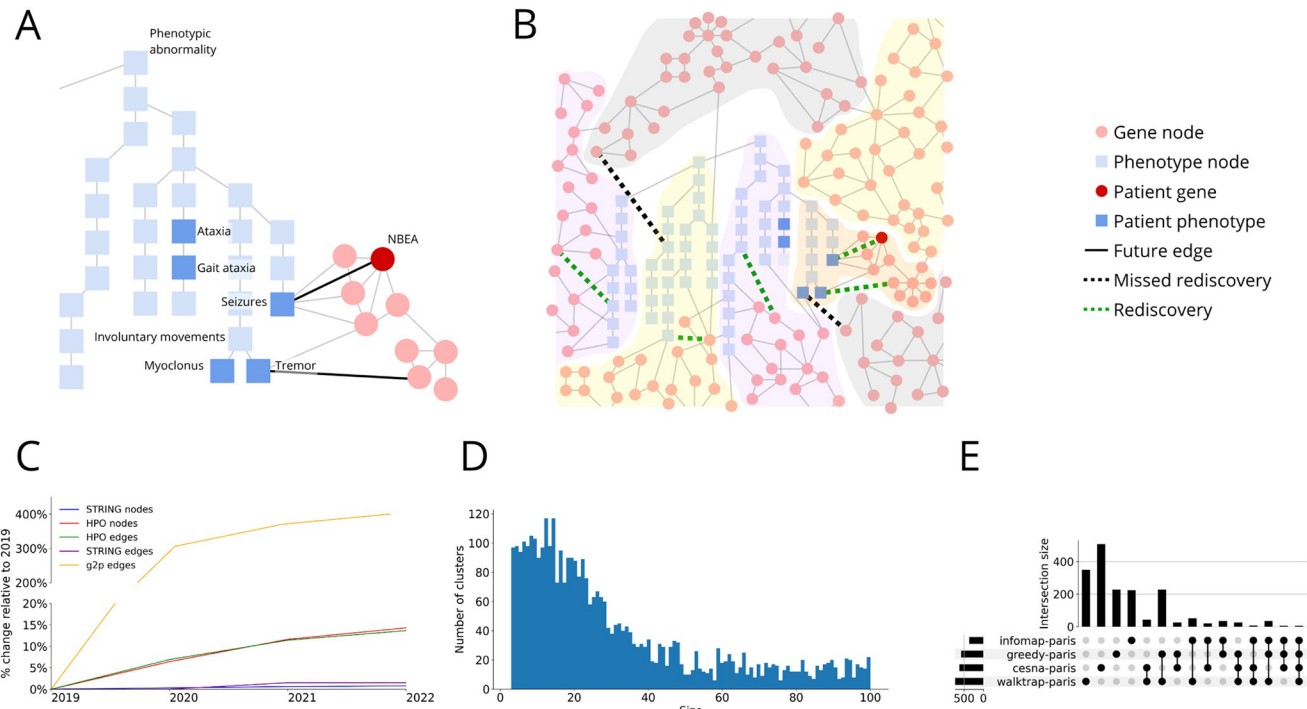

**Fig 1. Biological ontologies can be combined, clustered, and scored. A.** An example of how STRING (red circles) and HPO (blue squares) in 2019 could be combined into one large network. Phenotypes (dark blue squares) and variant harboring genes (dark red circles) for a MyGene2 patient are highlighted. Edge connecting nodes present in the 2019 network are thin and grey, and edges added in 2020 are bold and darkened. Note that with just the 2019 edges, *NBEA* and Seizures are not directly connected, but in 2020 there is an edge connecting them. **B.** The network is clustered, and rediscovery is performed and shown in an expanded view. Some clusters have genes and phenotypes, while others are homogeneous in node type. The edges added in 2020 are searched for in 2019 clusters; if both nodes of a new 2020 edge co-occur in the same 2019 cluster, it is a rediscovery (dotted green lines); otherwise, it is a miss (dotted black lines). The number of rediscoveries in a cluster can be compared to a null model to generate an empirical p-value for each cluster. **C.** Network growth by year. Percent growth of node and edge types in the graph relative to the amounts in 2019; mind the break in the y-axis. In 2019, there were 14,370 HPO terms, 19,536 genes, 18,199 phenotype-to-phenotype edges, 5,879,233 gene-to-gene edges and 42,079 g2p edges. HPO and STRING release regular updates; HPO has every 1–7 months since 2018, and STRING updates every 1–2 years. Both sources generally increase in size over time as new PPIs are discovered and new phenotypes are added to HPO. Over time, this growth enables us to evaluate how our methods would have performed in the past. **D.** Distribution of the size of the subclusters across all algorithms. **E.** The breakdown of rediscovered edges, commonality, and uniqueness is shown as an upset plot. A total of 1,299 different edges were rediscovered by a subcluster. 700 of these (54%) are unique—rediscovered by only a single algorithm. Unique percentages are 25.9% walktrap-paris with 347 edges total; 72.1% infomap-paris with 222 total; 0% greedy-paris with 226 edges total; 89.3% cesna-paris with 504 edges total.

It has been repeatedly observed and validated that gene products that interact with each other (e.g., have an edge between them in a PPI network) are more likely to share function [11]. This principle is called guilt-by-association and is the key assumption of virtually all network biology methods. Under this assumption, groups of nodes that are densely connected to each other in the network indicate some higher-order relationship between them, be it proteins belonging to the same pathway, genes pertaining to a complex disease module, or genes/proteins all targetable by the same drug. Such groups of nodes like this can be identified using network clustering algorithms.

Network clustering, also known as module detection or community detection, is a valuable technique in various fields, particularly in the analysis of complex systems like biological networks Fig 1B. Within biological networks, a variety of clustering algorithms can be used for many purposes such as disease module detection [12–16], identifying gene regulatory network [17–19] and identify functional modules [20].

Clustering algorithms are versatile, and the same algorithm is capable of answering questions in various situations. This is exemplified by similar classes of algorithms being applied

**Table 1. List of reviewed studies or tools related to clustering algorithms for biological networks.** This is nowhere near an exhaustive list of papers or tools on the topic. It is not intended to be a systematic review but highlights the breadth and general shift in the methods up to the present.

| Paper | Year | Algorithm | Network |
|---|---|---|---|
| [17] | 2008 | Weighted gene co-expression network analysis (WGCNA) | co-expression |
| [21] | 2009 | Markov, affinity propagation | PPI |
| [18] | 2010 | Self organized map | co-expression |
| [19] | 2010 | Modularity | co-expression |
| [15] | 2011 | Current flow | PPI |
| [22] | 2014 | Snowballing sampling | co-expression |
| [12] | 2015 | DIAMOND | Interactome—PPI and metabolite interactions |
| [23] | 2015 | Modularity | PPI |
| [24] | 2018 | Infomap | PPI |
| [14] | 2019 | Heavy subgraph detection—modularity | multi-layer: PPI, GO, DO, Disease symptoms profile connections |
| [13] | 2020 | Ensemble: co-expression clustering, bi-clustering, self-organizing map, k-means & co-expression | |
| [25] | 2020 | Ensemble: DIAMOnD [12], DiffCoEx [26], MCODE [27], MODA [28], ModuleDiscoverer [29] and WGCNA [17], | co-expression |
| [20] | 2021 | Markov | PPI |

repeatedly over the years in different studies and on different datasets (1). Thus, even though most of these studies are not focused solely on rare diseases, their ability to work on many biological networks opens the possibility of applying them to the study of virtually any disease or class thereof.

There is a trend toward using an ensemble of methods or methods capable of using higher-order patterns and integrating multiple types of networks (Table 1). In the early days of clustering biological networks, the trend was to use a single algorithm and a relatively small PPI network. Over time, many more types of algorithms were introduced; some were quickly abandoned, and others were used for a decade or more. In recent years, new tools have focused on using ensembles of popular tools of the past or new methods capable of leveraging higher-order patterns on their own.

But just because a method is old does not make it obsolete; one of the first clustering tools ever published, WGCNA [17], remains an extremely popular tool and was even included in a more recent ensemble method [25]. Markov models were considered old and were being replaced by affinity propagation in network science as a whole back in 2009, yet proved superior on PPI networks [21]. More recently, Markov models remain relevant due to their ability to use higher-order patterns, as [20] point out.

The desire for using more and higher order patterns is exemplified by paper since using ensemble methods [13, 25] and the return of Markov models [20].

Similar to the diversification of clustering algorithms, the underlying networks employed have grown in size and complexity. These networks are constantly being expanded, and as shown in Table 1, the networks used in these studies have slowly transitioned from simply PPI or co-expression networks to those of highly heterogeneous nature—multilayer networks and the logical next step knowledge graphs (KGs) as taken by [16].

We expand upon the work of clustering biological networks using an ensemble of clustering tools and a heterogeneous KG combined with a predictive model. To find new and meaningful clusters of genes and phenotypes in a densely connected network, we developed the biological ontology cluster classification (BOCC) tool. BOCC is a series of network-science-based methodologies that identify relevant clusters from a heterogeneous network comprised of HPO,

STRING, OMIM, and Orphanet. We take advantage of previous versions of these data sources to create a network time series by generating yearly snapshots of the network from 2019–2022. We apply a suite of network clustering methods to all. Then, in each cluster from $year_t$, we quantify the number of g2p edges added during $year_{t+1}$ where both nodes co-occur within that cluster and call these rediscoveries.

By comparing the number of rediscoveries in each cluster to a null model, we calculated an empirical p-value for each cluster. We deemed those with small p-values significant, containing notably more rediscoveries than expected by chance. Year after year, there were few significant clusters. To identify which present-day clusters were useful for future discovery, we needed a way to predict which clusters likely contained new g2p edges in the near future. We trained four XGBoost models on biological and network topology features to accomplish this. Each of these four models performed binary classification to predict if a cluster satisfies a certain p-value threshold: $p < 1.00$, $p < 0.35$, $p < 0.10$, and $p < 0.05$ (one model for each threshold). Area under the curve (AUC) on a held-out test set for these models was 0.82, 0.80, 0.75, and 0.71, respectively.

These clusters predicted to be significant by the models contained between 2.35 and 8.72 times as many edges with inferred existence based on known drug interactions and their insignificant counterparts. These clusters also contained g2p connections known in mice but not yet shown in humans. Working with the profiles of 721 patients with rare diseases from Children's Hospital Colorado, we found a significant number of potential novel g2p co-occurring pairs for 38 patient profiles. Additionally, BOCC provided potential g2p connections for 14 of 111 patients with no connection between their genes and phenotypes from the MyGene2 website. These co-occurring pairs are promising hypotheses for clinicians and researchers in these patients' diagnostic odysseys.

BOCC is freely available as a command line tool at https://github.com/MSBradshaw/BOCC and as an interactive web application at https://ryanlayerlab.github.io/BOCC/.

## Methods

BOCC aims to identify g2p associations that have not been documented in the literature but can be inferred through existing knowledge bases. The approach involves creating clusters in a hybrid network—which is a network with nodes representing multiple types of biological entities, genes/proteins, and phenotypes in this case—and looking for genes and phenotypes that co-cluster but are not directly connected (Fig 1B). Each co-clustering, unconnected gene/phenotype pair represents hypothetical g2p associations. To evaluate the plausibility of those associations, we use historical data to create a time series of networks from 2019 to 2022. The presence of future edges in past clusters compared to a conservative null model provides a powerful training dataset to model edge-rediscovery as a function of biological and network topology features for training our model. By analyzing clusters corresponding to a patient's medical history, BOCC can help generate new hypotheses about the architecture of their disease.

### Network construction

Our hybrid network combined the protein graph from STRING and the phenotype directed acyclic graph (DAG) from HPO using g2p edges from Orphanet and OMIM (Fig 1A). STRING has approximately 20,000 nodes that correspond to proteins/genes and approximately six million edges to represent their interactions. For example, STRING connects *NBEA* to *GRIN1*, *GRIN2B*, *DLG4*, *DLG3* and more, all of which are to participate in the assembly and cell surface presentation of NMDA receptors. HPO organizes $\tilde{1}3,000$ phenotype nodes into a

hierarchical, tree-like structure, where edges indicate an "is a" relationship so that phenotypes become more specific deeper in the tree. For example, the leaf node Nocturnal seizures has an edge pointing to the Seizures (indicating it is a type or subclass of seizure) node which itself is a distant child of the Phenotypic abnormality node (the root of the tree as utilized here). Edges connecting STRING and HPO originated from OMIM and Orphanet and are made available as annotations to HPO by The Monarch Initiative. These edges represent causal relationships. For example, Seizures are known to be caused by a variant in *GRIN2B*, thus there is an edge between the two nodes. Prior to constructing the network, we removed all non-phenotypic-abnormality nodes from HPO. The full network contained non-phenotype hub nodes such as Autosomal recessive inheritance (HP:0000007) and Autosomal dominant inheritance (HP:0000006) that drastically reduced the distance between many nodes of the graph, making it difficult to infer phenotype similarity from the network topology.

Network change and growth: These data sources are continuously growing as new discoveries are made, leading to the addition and removal of edges and nodes in the network. An example of the change in size of HPO and STRING from 2019 to 2022 is shown in Fig 1B. Historical data about these biological networks can be used to create dynamic networks—a network whose structure changes over time. For example, the 2019 versions of STRING, HPO, OMIM, and Orphannet produced a 2019 network. This process was repeated for each year from 2019 to 2022 (S1 Methods). Prior to 2019, the recording and versioning of HPO and the g2p files were unreliable. The study of dynamic networks is fairly recent [30] and its application in biology is yet to be seen though ripe with applications [31]. One advantage of using a dynamic network is that when it comes to graph learning and link prediction a dynamic network creates a more realistic evaluation by capturing the temporal dynamics and dependencies present in biological systems. By incorporating the time series of the network for training and evaluation the models can leverage the knowledge of the recent past and its future to make predictions, mimicking the real-world scenarios where network structure evolves over time. This realistic evaluation allows for a more accurate assessment of the predictive models' performance in biological networks.

## Clustering

Clustering has several advantages. First, clusters are interpretable. The algorithms are simple and mathematically explainable, and co-clustered nodes offer some detail as to how the unconnected genes and phenotypes may relate. Second, clustering is unsupervised. The limited information available about rare and undiagnosed diseases makes it nearly very difficult to create a labeled training dataset that is large enough to be helpful for most machine-learning methods. Clustering does not require labeled data and can find new patterns among human physiology and disease networks and ontologies.

We used four clustering methods to identify related, but not necessarily connected, genes and phenotypes. These methods were selected because they were based on different algorithmic approaches that included greedy modularity maximization that maximizes dense connections between the nodes [32], walktrap which uses a random-walks-with-restart [33], infomap which is based on ideas of information theory [34, 35], and CESNA which forms communities from edge structure and node attributes [36] (S1 Fig, S1 Table) (see S1 Methods).

Our goal was to find sets of genes and HPO terms that were useful to clinical researchers, but these clustering algorithms identified clusters that were too large to be comprehensible (e.g. infomap identifies a cluster with greater than 20,000 nodes)(S2 Fig). Following discussion with clinicians, we determined that the clusters should not have more than 100 nodes, which agrees with the Disease Module Identification DREAM Challenge [37]. To bound the size, we

re-clustered all previously identified clusters using Paris hierarchical clustering [38], specifying an upper limit to cluster size. The result was many more small clusters (Fig 1D).

## Edge-rediscovery

Every unconnected co-clustered gene/phenotype pair represents a potentially latent g2p connection. To validate the biological relevance of these connections, we used our hybrid network time series to count the potential connections in one year's network that were confirmed by the subsequent year's network. For example, after creating two distinct networks and clusters from the 2019 and 2020 versions for STRING, HPO, OMIM, and Orphanet and performing clustering on the 2019 network, there were 128,798 new edges in the 2020 network that did not exist in 2019 (but both nodes were present). These edges represent the g2p discoveries made in 2020. After performing clustering on the 2019 network, we found of the 2020 g2p edges, 696 had co-occurring vertices in our 2019 clusters, or "rediscoveries" as we like to call them.

It is worth noting that our rediscovery method had limitations. First, our definition of rediscovery was conservative and only considered findings made in the following year. Just because a cluster had no rediscoveries in year $t + 1$, does not mean it would not have any in $t+ 2$ or $t+ 3$; year after year, it becomes more probable that a cluster has a rediscovery (Fig 2). For example, there were 696 rediscoveries from 2019 to 2020 and 1011 from 2019 to 2021. Using a short 1-year peek into the future, as few as 15% of clusters had one or more rediscoveries. If we look 3 years into the future, as many as 37% of clusters had a rediscovery. Second, the number of rediscoveries is bounded by the priorities and the productivity of the biomedical research community. For example, the number of g2p edges grew by a factor of 3 from 2019 to 2020, which is likely explained by a burst in curation efforts in the Monarch Initiative. Regardless of the clusters BOCC creates, due to the incomplete and ever-advancing nature of biomedical knowledge, many of its unconnected co-clustered vertices would remain undiscovered for years, and given the technical and ethical limitations of research, we would expect that many of the connections would never be found.

Each clustering algorithm rediscovered different information. Consider the specific edges rediscovered from 2021 in the 2020 clusters. In total, there were 31,858 edges that could possibly be rediscovered. Across the four algorithms, 1,299 total edges were rediscovered. 599 were rediscovered by two or more algorithms, and the rediscovery of 700 was unique to a single algorithm. While one algorithm may outperform another in the total number of edges rediscovered, the high amount of edges rediscovered by only a single algorithm suggested we should not choose a "best" algorithm, but instead consider them all in-concert for the best results (Fig 1E). This coincides with the findings of Peel et al. 2017 [39] that clustering algorithms abide by the idea of No Free Lunch—none are universally optimal.

Given the size and complexity of our hybrid network, rediscovered edges could occur by chance. To evaluate how likely this is for a given cluster, we compared the number of rediscoveries per cluster to the expected number using a snowball-sampling-inspired [40] null model (Fig 3A–3F). Our null model generated a set of 10,000 synthetic clusters by selecting one node at random from the network, then iteratively adding that node's neighbors, those neighbor's neighbors, and so on until the size of the original cluster was reached, and then repeated this process 10,000 times. We then performed edge-rediscovery on the synthetic clusters as described earlier. This results in an empirical rediscovery distribution from which we could extract the expected number of rediscovers and determine the significance of the observed rediscovery count. We chose this method over a random-graph or edge-shuffle null model because the random-graph model tended to overestimate significance in our network (S3 Fig)

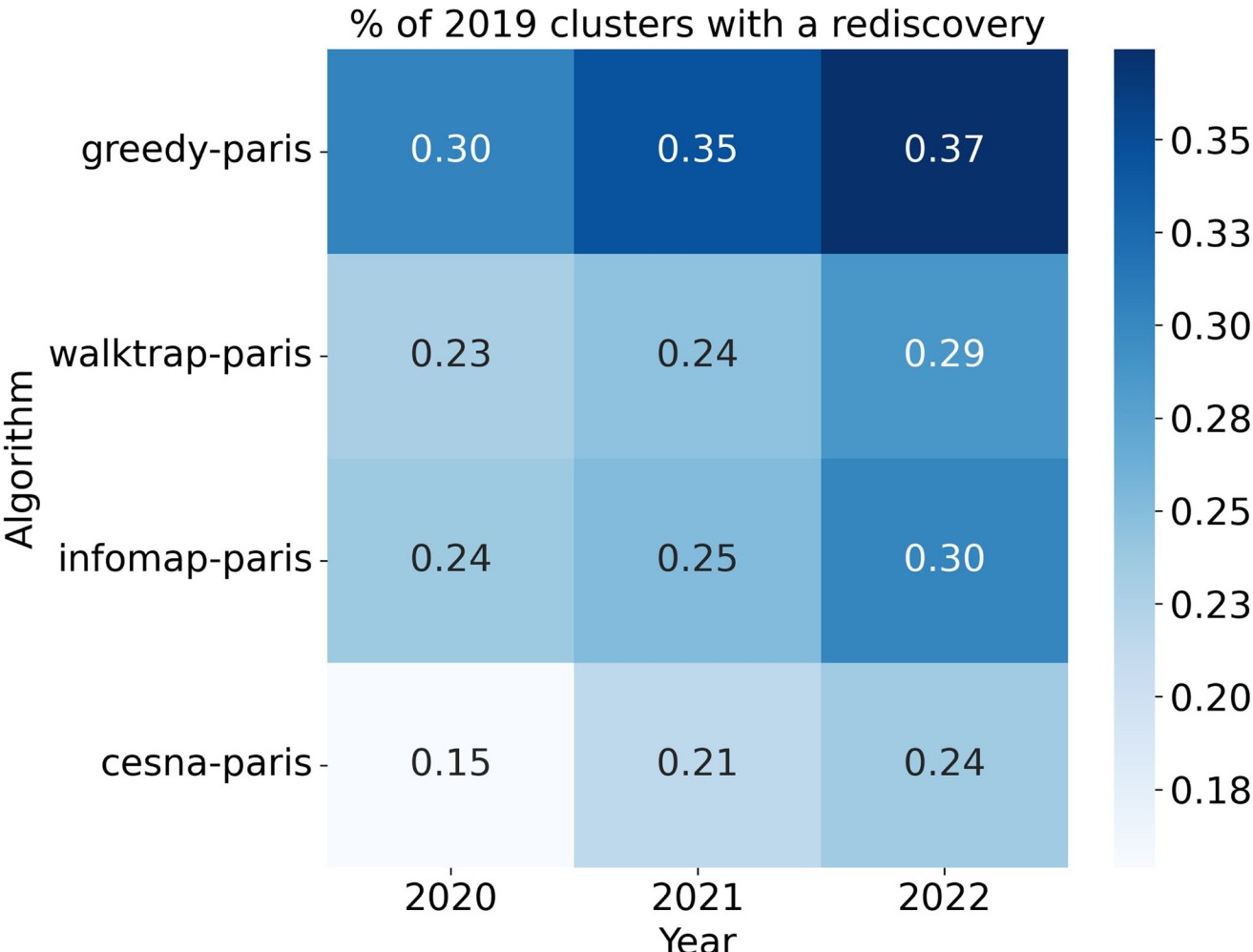

**Fig 2. Effect of time on the proportion of 2019 clusters with at least one rediscovery, when rediscovery is performed with the new edges added in successively later years, 2020–2022.**

and biological networks in general [12, 23]. Among the 41,968 2019 clusters, 1,323 are non-trivial (heterogeneous and have > 3 members). Of these non-trivial clusters, 255 have at least one rediscovery (19.3% of non-trivial clusters). The distribution of the empirical p-values of these non-trivial clusters with at least 1 rediscovery can be seen in Fig 3G–3J.

## Predicting cluster usefulness

Using historical data, we identified many clusters, only a portion of which contained rediscoveries in the subsequent year and a smaller portion of which had more rediscoveries than expected under the null model. We anticipated this observation would continue to be true in the future, which necessitated a means of predicting which clusters were most likely to contain a substantial number of rediscoveries in the near future. To do this we trained an XGBoost classifier [41] with a DART model (Dropouts meet Multiple Additive Regression) [42] as the booster to predict if a cluster satisfies several significance thresholds (Fig 4A). Due to the high amount of variation in the predictive performance across significance thresholds, rather than selecting a single threshold, we selected several to allow users to

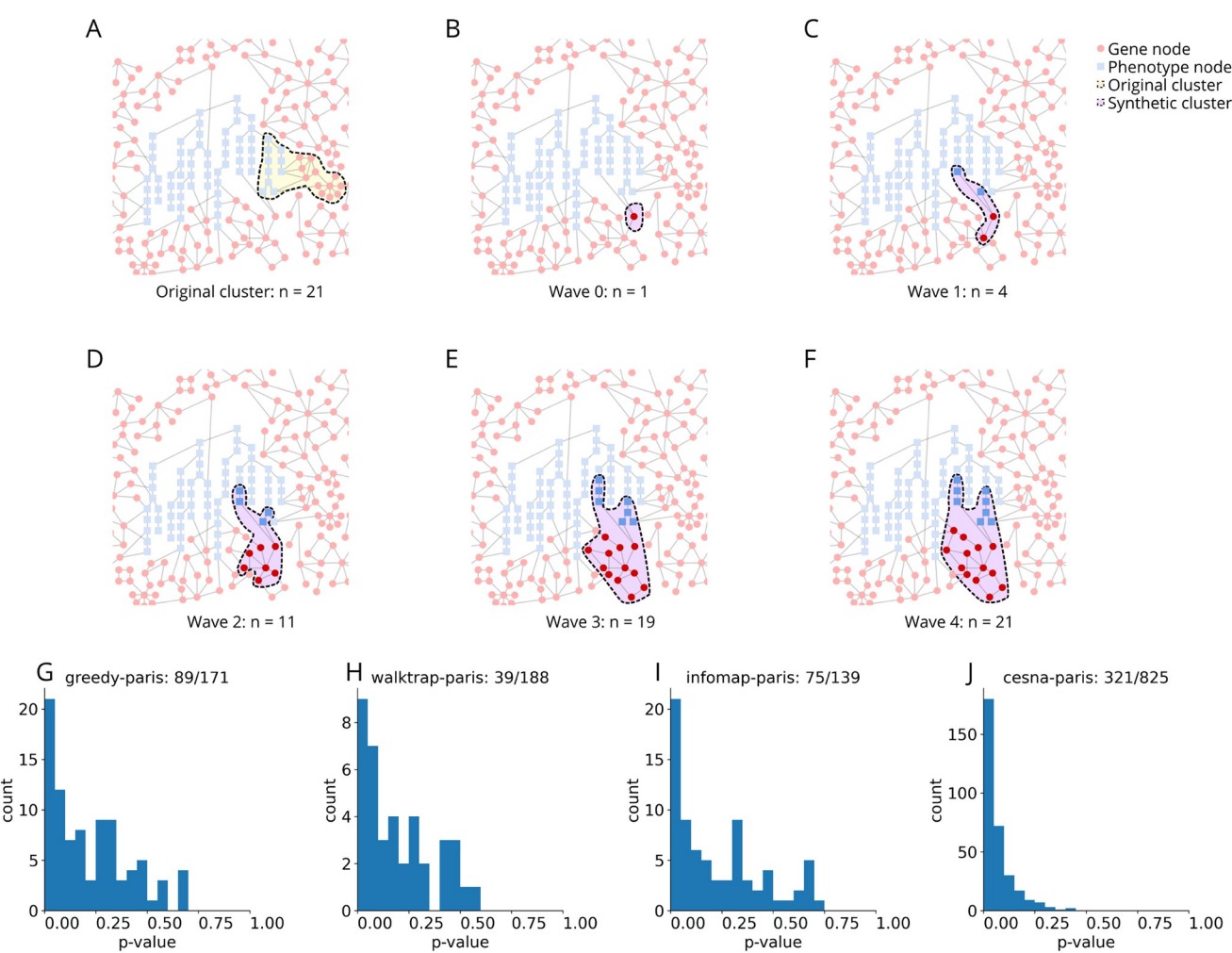

**Fig 3. Graphical representation of the snowball sampling process.** Snowball sampling **A.** starts with a real cluster identified in the STRING-HPO network and makes note of its original size (n = 21). Then we start to build a synthetic cluster by **B.** picking a starting node from the network at random and adding it to a synthetic cluster. **C.** Grow the synthetic cluster by adding the neighbors of nodes in the synthetic cluster. **D—E.** Continue the growth by adding neighbors of the nodes in the synthetic cluster and those neighbors' neighbors. **F.** When adding all of the neighbors of nodes in the synthetic cluster would exceed the size of the original cluster (21), choose neighbors at random to finish filling the synthetic cluster. This process is repeated 10,000 times for each cluster, then the number of rediscoveries in the real cluster is compared to that of the 10,000 synthetic clusters to calculate an empirical p-value. **G-J** Distribution of empirical p-values based on 10,000 synthetic clusters generated via snowball sampling. Only the p-value of clusters that had at least one rediscovery is shown. **G.** Greedy-paris, there were a total of 171 non-trivial clusters, 89 of which had at least one rediscovery. **H.** Walktrap-paris, there were a total of 188 non-trivial clusters, 39 of which had at least one rediscovery. **I.** Infomap-paris, there were a total of 139 non-trivial clusters, 75 of which had at least one rediscovery. **J.** Cesna-paris, there were a total of 825 non-trivial clusters, 321 of which had at least one rediscovery.

choose the balance between predictive performance and the likelihood of clusters containing a non-random abundance of new edges in the near future. Specifically, we train four models to predict if clusters' p-values are $p < 1.00$, $p < 0.35$, $p < 0.10$, or $p < 0.05$. Analysis of ROC AUC of all thresholds up to 1.00, in .05 steps, revealed these four thresholds led to the best model performances (Fig 4B). To address the imbalance of classes (if a cluster's p-value is above or below the threshold) when thresholding the p-values for training, we randomly down-sampled the majority class without replacement to equal the size of the minority class. This resulted in 779, 700, 496, and 343 samples of each class for the models trained with thresholds $p < 1.00, 0.35, 0.10, 0.05$ respectively.

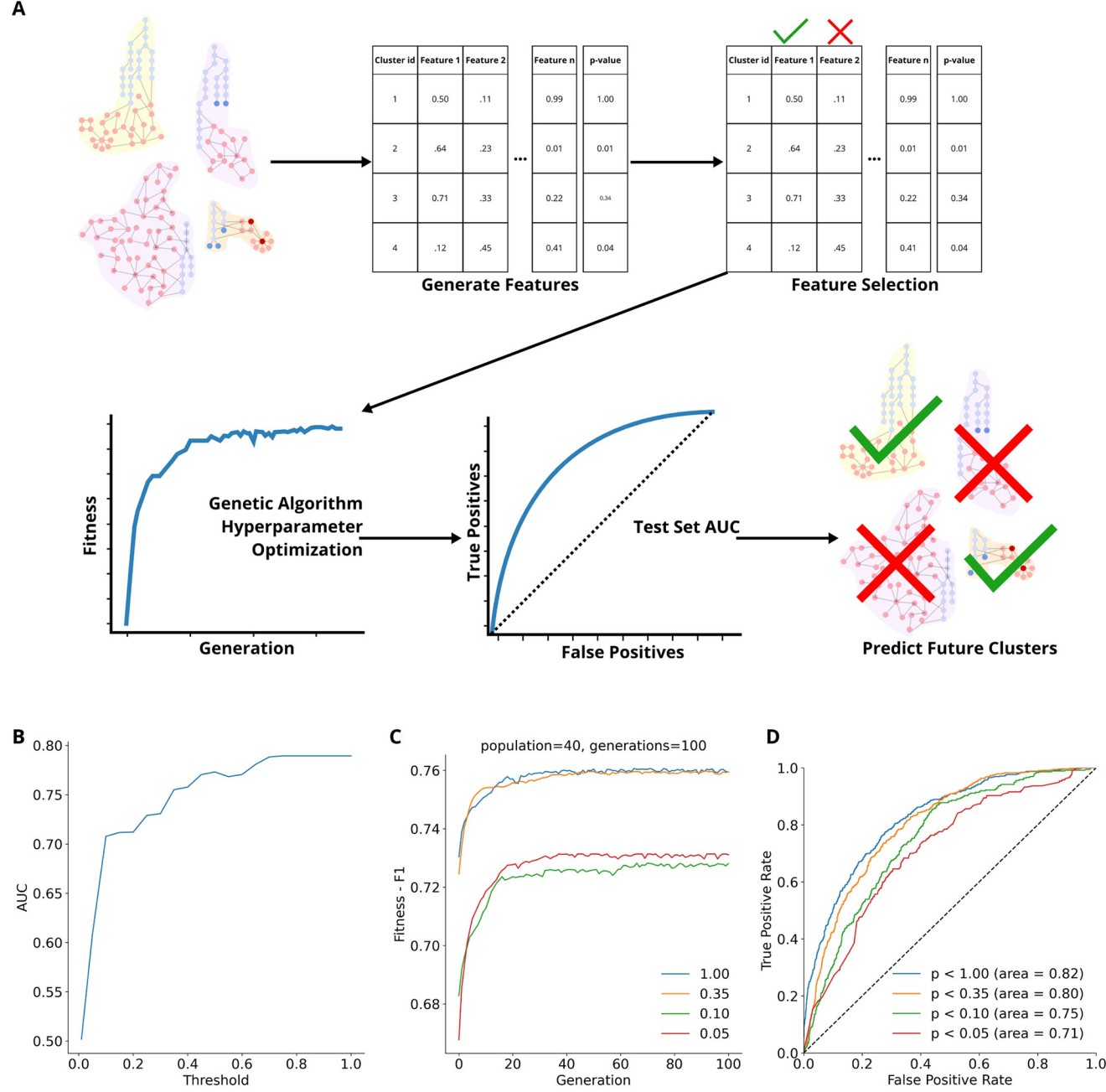

**Fig 4. A.** Procedure for training and evaluating the XGboost model. Features were generated about each cluster (identified as described in Section) describing them with a series of biological and network topological features, following which feature selection was performed. Hyperparameters were tuned using a genetic optimization algorithm. Final trained models are evaluated on a held-out test set. Models were then used to make predictions about clusters from the most up-to-date version of the network. **B.** AUC model performance as a function of p-value threshold. For each value across the x-axis, we trained a model to predict $p < threshold$ and reported the validation set AUC of that experiment on the y-axis. Hyperparameters used in this analysis were those originally established for $p < 1.00$. All models were trained on the 2019 clusters and validation was done on the 2020 clusters. Notice that the AUC has inflection points, and stops its origin rapid increase, at $p = 0.10$ (AUC = 0.71) and $p = 0.35$ (AUC = 0.76) achieving performance similar to $p < 1.00$ (AUC = 0.79) despite being substantially more stringent. **C.** Change in fitness (F1 score) as the genetic algorithm optimizer progressed. **D.** Hold out test set ROC curves for the models trained to predict the four different thresholds of significance. The dashed black line marks a line with a slope of 1 and AUC = 0.50.

**Table 2. Cluster features used as input to the XGBoost model.** The features describe both the biological and topological aspects of the clusters. Brief descriptions and the mean absolute SHAP value for variable importance are also provided for each feature. Ranges are based on those observed in the 2019 & 2020 clusters.

| Feature Name | Description | Range | SHAP |
|---|---|---|---|
| Size | Number of nodes in the cluster | 3–100 | 0.3 |
| Gene ratio | Number of genes in a cluster divided by cluster size. | 0.01–0.99 | 0.18 |
| Disease specificity | Proportion of genes participating in disease enrichment. | 0.00–1.00 | 0.07 |
| Edges inside | Number of edges internal to the community. | 2–2711 | 0.07 |
| AID | Average internal degree, defined as the average degree of all nodes in cluster. | 1.3–58.3 | 0.02 |
| NGM | Newman-Girvan modularity The number of internal edges minus the expected number of internal edges where edges are lain randomly to preserve the degree of each node (Newman, 2006). | 0.99–1.00 | 0.02 |
| AE | Average embeddedness of all nodes in cluster where the embeddedness of a node is its internal degree with respect to its overall degree. | 0.00–0.96 | 0.02 |
| Conductance | Time required for a random walk on the cluster to achieve its stationary distribution. | 0.17–1.00 | 0.01 |
| Cut | Defined as the fraction of existing edges leaving the community. | 0.00–0.08 | 0.01 |

**Table 3. The hyperparameters and their search ranges used by the genetic algorithm, final values for each threshold model are listed in their given columns.**

| Name | Range | $p < 1.00$ | $p < 0.35$ | $p < 0.10$ | $p < 0.05$ |
|---|---|---|---|---|---|
| learning_rate | 0.001–0.1 | 0.056 | 0.014 | 0.007 | 0.014 |
| gamma | 0.01–1.00 | 0.469 | 0.926 | 0.887 | 0.805 |
| n_estimators | 1–500 | 55 | 33 | 15 | 102 |
| max_depth | 1–30 | 4 | 5 | 9 | 1 |
| max_leaves | 1–10 | 8 | 6 | 2 | 10 |
| subsample | 0.01–1.00 | 0.122 | 0.315 | 0.642 | 0.650 |

For each subcluster, we calculated a series of 20 biological and network topology-based features. Feature selection was performed using the greedy Sequential Feature Selection method [43] which resulted in a set of 9 features (Table 2). Feature importance was determined using SHAP (SHapley Additive exPlanations) [44], and the model for predicting clusters with $p < 0.35$.

We tuned the hyper-parameter for each model using a genetic algorithm optimizer [45]. We used a population of 40, 100 generations, and F1 score as the fitness metric. A full list of the hyperparameters search space and the final values for each threshold model can be seen in Table 3. The fitness improvement over time can be seen in Fig 4C.

Our final models were trained using the selected features (Table 2), their corresponding hyperparameters (Table 3), and class-balanced sets of clusters from 2019 and 2020. As our test set, we use the 2021 clusters. The models achieved ROC AUC of 0.82, 0.80, 0.75, and 0.71 for the models trained to predict $p < 1.00$, 0.35, 0.10, or 0.05 (Fig 4D).

## Processing clinical samples

We performed a retrospective study of medical records and used BOCC to generate hypotheses for 721 fully anonymized rare and undiagnosed disease patient profiles at Children's Hospital Colorado (CHCO). CHCO granted permission to use the data, and the Colorado Multiple Institutional Review Board (COMIRB) approved the protocol used here. Records were accessed on November 8th 2023, and authors did not have access to any identifiable information. These patients had all previously undergone whole exome sequencing (WES) and had their conditions described with HPO terms by clinicians. Patients had between 1 and

17 associated HPO terms (median 3). For our analysis, we used the variant call format (VCF) files generated by CHCO's standard variant calling pipeline, which identified, on average 42,449 SNPs and indels in each patient. These variants were filtered down to an average of 401 variants per patient by selecting variants that met the following criteria:

- $f$ = population frequency (0.00–1.00)

- $i$ = is insufficient population frequency (true/false)

- $e$ = is located on an exon (true/false)

- $s$ = is located on a splice site (true/false)

- $c$ = is a coding effect other than synonymous (true/false)

- $l$ = is labeled pathogenic/likely-pathogenic (P/LP) based on ClinVar scoring (true/false)

- select variant if $(f < 0.01 \parallel i)$ & $(e \parallel s)$ & $(c \parallel l)$

Given the number of patients, affected genes, and HPO terms we searched for, there was a real possibility of false positive co-occurring pairs. To control for this, we generated empirical p-values for each patient by comparing the number of co-occurring gene and phenotype pairs each patient had to the number of co-occurring pairs expected based on a null model. Our null model operated by doing the following for each patient:

1. Choose a set of HPO terms belonging to another patient at random from the set of all patients.

2. Count co-occurring pairs found between the patient's affected genes and the random set of HPO terms.

3. Repeat steps 1 and 2 100 times.

4. Calculate the empirical probability the number of co-occurring pairs under the null model is greater than or equal to the observed.

## BOCC based tools

To use BOCC to generate hypotheses for a specific patient, one can search the clusters for co-occurring pairs of an affected gene and presenting phenotype in the clusters predicted to be significant. We provide two tools for accessing and interacting with BOCC: a web-based network visualization tool and a command-line interface (CLI). Access the visualization tool at https://ryanlayerlab.github.io/BOCC/.

The BOCC CLI is better suited for searching for many g2p connections in the clusters. The requirements for the CLI and full documentation can be found at https://github.com/MSBradshaw/BOCC.

The BOCC web app allows users to search for HPO terms and genes within the BOCC clusters in a browser-based interactive manner. Once a cluster has been selected, it is easy to see how genes and HPO that are not directly connected could be. A screenshot of the web app exploring one of the MyGene2 patients can be seen in Fig 5. To use the BOCC web app for hypothesis generation, a researcher can search for a series of HPO terms and genes, and then view clusters where the search terms co-occur in the context of other relevant genes and HPO terms. All clusters predicted to exceed any threshold of significance will be reported and ordered according to the number of matching terms, followed by the predicted significance threshold, ties in the order are arbitrarily broken alphabetic order of the cluster-ID.

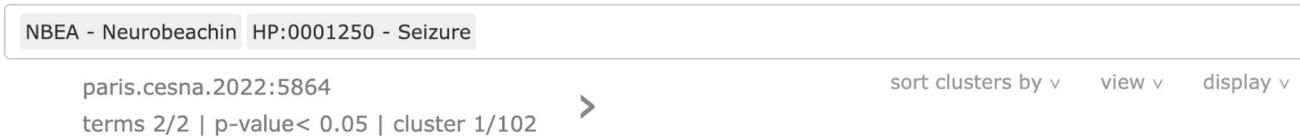

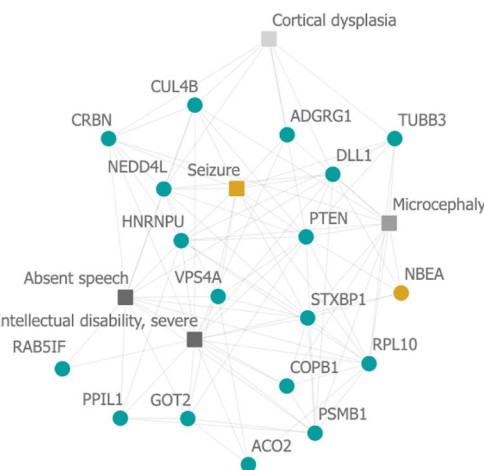

**Fig 5. Screenshot from the BOCC visualization web app.** An example is shown for MyGene2 patient 1930, searching for one of their affected genes *NBEA* and the HPO term Seizures (HP:0001250), both highlighted as the only yellow nodes. Phenotype terms are shown as square nodes and genes as circles. The darkness of the phenotype terms is indicative of the depth of the term on the HPO tree—a proxy for how specific the term is. *NBEA* and Seizures have no direct connection between them, however, they co-occur in a cluster that the $p < 0.05$ model predicted to contain a significant number of discoveries in the near future. In this cluster, they share numerous neighbors that are fairly densely connected to each other.

## Results

### CHCO patients

We applied BOCC to 721 patients from Children's Hospital Colorado (CHCO) with suspected genetic drivers of their diseases. These patients had all previously undergone whole exome sequencing (WES) and had their conditions described with HPO terms by clinicians. Patients had between 1 and 17 associated HPO terms (median 3) (Fig 6A). For our analysis, we used the variant call format (VCF) files generated by CHCO's standard rare disease variant calling pipeline, which identified, on average, 42,449 SNPs and indels in each patient (Fig 6B). After performing quality control (Section), there were, on average, 401 variants per patient (Fig 6C). We then used the BOCC CLI to search for co-occurrences of each patient's potentially affected genes and HPO terms.

BOCC found potential co-occurring pairs for 619 patients in clusters predicted to be significant by the $p \leq 0.05$ model, (Fig 6D). From this experiment, we found that 38 of the 619 patients with a co-occurring pair had significantly ($p \leq 0.05$) more co-occurring pairs than expected by the null model as described in Section (Fig 6E). Similar results were found when using clusters predicted by models trained on the thresholds $p \leq 1.00, 0.35, 0.10$ (Table 4).

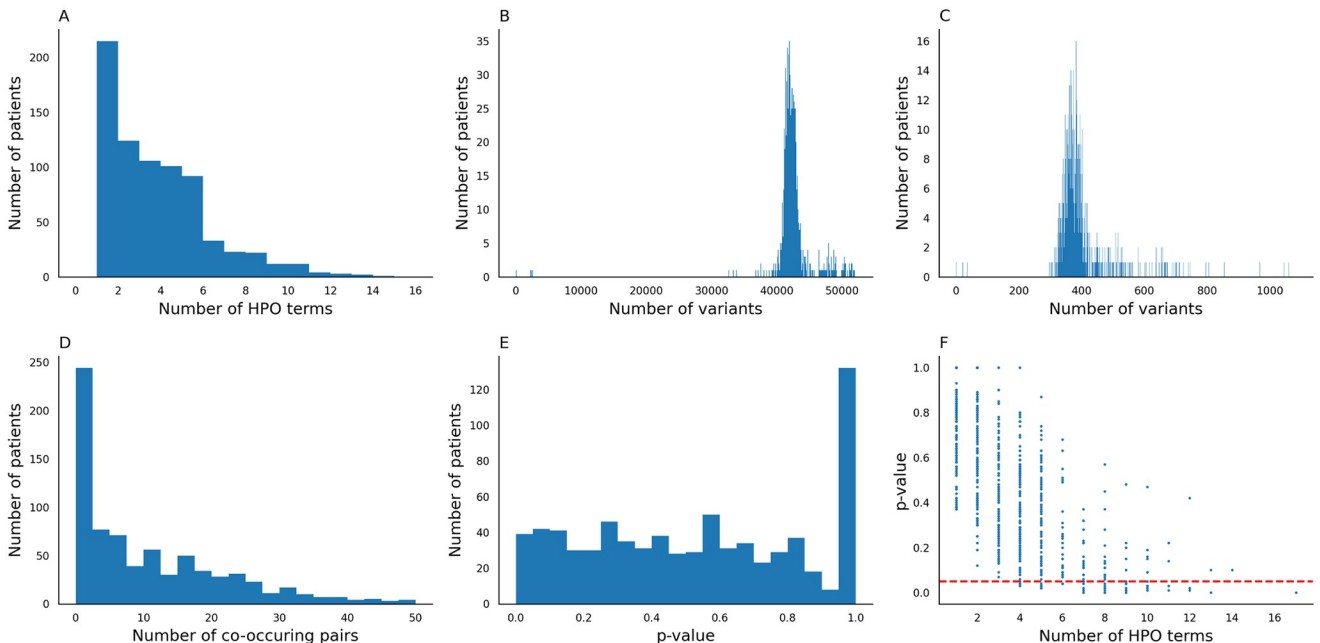

**Fig 6. A.** Distribution of the number of HPO terms associated with each patient. **B.** Distribution of the number of records in each VCF after going through CHCO's in-house quality control filtering. **C.** Distribution of the number of variants in each patient's VCF. **D.** Distribution of the number of g2p co-occurring pairs found for each patient. **E.** Distribution of the empirical p-values calculated for the patients using the null model centered around shuffling the randomly shuffling sets of HPO terms. **F.** The relationship between the number of HPO terms assigned to a patient and their p-value. Including all points, there is a Pearson's correlation coefficient R=−0.71, considering only point $p < 1.00$, R=−0.66.

One may expect the p-value to directly result from the number of HPO terms assigned to each patient. We did not find this to be the case. While there is evidence of some correlation between the two values (Fig 6F), there were still a substantial number (21) of samples with significant p-values with fewer than 8 HPO terms, with a great degree of variability within the lower numbers of HPO term samples. Similarly, having a high number of HPO terms does not guarantee a low p-value; for example, the patient with the second largest number of HPO terms, 14, only had a p-value of 0.10.

## MyGene2

Having validated BOCC's ability to rediscover information on a rigorous null model and validation and test sets based on real times series (see Figs 3 and 4), we then applied it to undiagnosed disease patient data found on MyGene2 (https://www.mygene2.org/MyGene2/) [1]. MyGene2 is a website dedicated to helping families, clinicians, and researchers connected to

**Table 4. Number of g2p co-occurring pairs in BOCC clusters and the number of those patients whose number of co-occurring pairs is significant compared to the HPO list shuffle null model.** Breakdowns are given for the four sets of clusters identified by the corresponding predictive models.

|  | Num. with match | Num. significant |
|---|---|---|
| Model $p \leq 0.05$ | 619 | 39 |
| Model $p \leq 0.10$ | 612 | 35 |
| Model $p \leq 0.35$ | 620 | 32 |
| Model $p \leq 1.00$ | 603 | 35 |

**Table 5. List of MyGene2 patients and g2p pairs found co-occurring in a 2022 BOCC cluster and prediction results from the models trained for four different thresholds.** The final column denotes the model with the lower threshold that predicted the cluster to be significant.

| Patient | Cluster-Id | Gene | HPO Term Name | Best level of prediction |
|---------|-----------|------|---------------|--------------------------|
| 1292 | paris.walktrap.2022:1369 | *TNNT3* | Arthrogryposis multiplex congenita | $p < 0.05$ |
| 151 | paris.infomap.2022:973 | *PCOLCE2* | Myopia | $p < 0.05$ |
| 1930 | paris.cesna.2022:5864 | *NBEA* | Seizure | $p < 0.05$ |
| 2197 | paris.cesna.2022:17456 | *C17orf62* | Fever | $p < 0.05$ |
| 2197 | paris.cesna.2022:17456 | *C17orf62* | Recurrent pneumonia | $p < 0.05$ |
| 2197 | paris.cesna.2022:17456 | *C17orf62* | Splenomegaly | $p < 0.05$ |
| 2197 | paris.cesna.2022:17456 | *C17orf62* | Lymphadenopathy | $p < 0.05$ |
| 2234 | paris.cesna.2022:15933 | *DROSHA* | Microcephaly | $p < 0.05$ |
| 2525 | paris.cesna.2022:5117 | *FMN1* | Short stature | $p < 0.05$ |
| 2584 | paris.cesna.2022:4308 | *PYROXD1* | Myopathy | $p < 0.05$ |
| 2649 | paris.walktrap.2022:1626 | *BCL6B* | Alopecia | $p < 0.05$ |
| 27 | paris.cesna.2022:3984 | *SCN2A* | Seizure | $p < 0.05$ |
| 2748 | paris.cesna.2022:18491 | *DROSHA* | Dysphagia | $p < 0.05$ |
| 347 | paris.cesna.2022:11079 | *GUCY1A3* | Hypertension | $p < 0.05$ |
| 8 | paris.cesna.2022:2747 | *ACTC1* | Low-set ears | $p < 0.05$ |
| 8 | paris.cesna.2022:13667 | *ACTC1* | Short stature | $p < 0.05$ |
| 8 | paris.cesna.2022:2747 | *ACTC1* | Ventricular septal defect | $p < 0.05$ |
| 877 | paris.cesna.2022:11542 | *B9D2* | Ataxia | $p < 0.05$ |
| 878 | paris.cesna.2022:11542 | *B9D2* | Ataxia | $p < 0.05$ |

individuals living with a rare or undiagnosed disease with similar genotypic and phenotypic information, which can go a long way to help the n-of-1 problem. MyGene2 provides a location on the internet for users to upload and publicize genetic information, lists of phenotype HPO terms, and patient background information. The amount of information and degree of privacy is determined by the users. Some users upload many paragraphs of background information and photos of the patient accompanied by a VCF and a long list of HPO terms, which are all publicly available. Other users opt to upload only a single candidate gene/variant and a few HPO terms that are only accessible to those who have also contributed to MyGene2 (other families, clinicians, or researchers). We scraped all publicly accessible profiles on MyGene2 (912 in total), (S1 Dataset). We found that presently 111 of these profiles contained no direct connection between any of their genes and any of their HPO terms; we assumed these cases to still be undiagnosed. We then searched for co-occurrences of g2p pairs related to their undiagnosed profiles in our clusters from the 2022 network. We found 19 pairs from 14 MyGene2 patients where the currently unconnected gene and phenotype co-occurred in the same cluster (Table 5).

Of these MyGene2 patients' gene-phenotype pairs found in BOCC clusters, all shortest paths between the two nodes have only a single intermediary gene node connecting them.

## Mammalian phenotype ontology

We further evaluated the utility of the clusters using g2p connections present in mice but not present in our 2022 network. We identified 90 g2p connections known from knock-out studies in mice but not yet observed in humans. The clusters suggest that 5 of these pairs may yet exist in humans (Table 6). One of these pairs (*SLC4A1* and HP:0001927) had supporting evidence coming from multiple clusters. These 90 theorized connections we searched for were made by

**Table 6. g2p edges inferred from mice found co-occuring in a BOCC cluster.** The final column denotes the model with the lower threshold that predicted the cluster to be significant.

| Cluster-Id | Gene | HPO Term Name | Best level of prediction |
|---|---|---|---|
| paris.walktrap.2022:307 | SLC4A1 | Acanthocytosis | $p < 0.05$ |
| paris.infomap.2022:994 | SLC4A1 | Acanthocytosis | $p < 0.05$ |
| paris.cesna.2022:16324 | IDS | Hepatomegaly | $p < 0.05$ |
| paris.cesna.2022:6103 | MSH2 | Astrocytoma | $p < 0.05$ |
| paris.infomap.2022:1131 | FANCD2 | Ectopic kidney | $p < 0.05$ |

connecting Mammalian Phenotype Ontology (MPO) and their mouse gene knock-out studies to human resources going from MPO → OMIM → HPO. The MPO → OMIM connections were made by Sardana et al 2010 [46] and were gathered from their S1 Dataset.

## Rediscovery of drug inferred edges

We created a list of g2p edges that do not exist in humans but are inferred based on drug-to-gene and drug-to-disease edges from the Comparative Toxicogenomics Database (CTD) [47]. CTD identifies 101,300,249 potentially latent gene-to-disease connections using drugs as intermediaries between genes and disease. Many of the substances in CTD are listed as both therapeutic for a disease and causative of the same disease, we remove all such edges (97,795,120). We further removed an additional 2,233,706 edges that did not have a mapping from their disease MESH ID to an HPO term. This left us with 1,147,313 inferred g2p edges. We searched for these edges in clusters and found substantially more edges occurring within clusters predicted significant than not (Table 7). There were between 2.35 and 8.72 times as many discovered edges in clusters predicted significant versus not—further evidence that our models can identify which clusters are enriched for latent g2p connections.

## Discussion

Rare diseases affect 25 to 30-million people in the United States [48]. Despite this, it is not uncommon for diagnostic odysseys to last 4–8 years, with many patients never receiving any final diagnosis [2–5]. We have presented BOCC, an analytical pipeline comprised of method-ologies and tools for hypothesis generation and exploration. We demonstrated its utility for proposing novel gene-to-phenotypes connections in 38 patient profiles provided to us by Chil-dren's Hospital Colorado and on a total of 14 patients publicly listed on MyGene2.

Ultimately BOCC is an aid to clinicians still searching for answers after standard diagnostic and variant prioritization methods have failed to yield conclusive results. BOCC is not a vari-ant prioritizer; it is a tool intended to cast a wide net by expanding the search space and con-sidering known relationships and interactions of genes and phenotypes. To reduce the chance of false positives, we recommend limiting the searches with BOCC to genes harboring variants that have already been prioritized by another tool or have other preliminary evidence

**Table 7. Number of CTD inferred g2p edges discovered with cluster predicted to be significant vs non-significant based on the four thresholds.**

| | <1.00 | <0.35 | <0.10 | <0.05 |
|---|---|---|---|---|
| Sig. | 3619 | 3581 | 2830 | 3281 |
| Not Sig. | 415 | 453 | 1204 | 753 |

suggesting their culpability but lacked sufficient evidence on their own—such as the quality control process used with the CHCO patients.

In addition to these more specific contributions, we have learned many things along the way that are more broadly applicable to users of biological networks. The use of historical data to create a network time series is a relatively new development in network science, even more so in applications to human health, and presents advantages when training predictive models on networks. Frequently, when building predictive models that operate on biological networks "a fraction of links from the current graph structure is deleted, and taken as the test set" because "one cannot know the future links of a graph at time" [49]. While it is true we do not know the future, we do however know the past and the past's future, which we used here for testing and training. Here, we created and used time-stamped versions of our network in the training and evaluation of our models. This is in essence a dynamic network—a network whose structure changes over time. The study of dynamic networks is fairly recent [30] and its application in biological is yet to be seen, though ripe with applications [31]. One advantage of using a dynamic network is that it creates a more realistic evaluation by capturing the temporal dynamics and dependencies present in biological systems. By incorporating the time series of the network for training and evaluation, the models can leverage the knowledge of the recent past and its future to make predictions, mimicking the real-world scenarios where network structure evolves over time. This realistic evaluation allows for a more accurate assessment of the predictive models' performance in biological networks.

We found the structure of the network itself must be carefully considered. When constructing our relatively small knowledge graph from STRING and HPO, we found that certain terms in HPO were extremely highly connected (e.i. hub nodes) and not necessarily informative in predicting novel g2p connections. This includes nodes related to the sub-trees of HPO related to "Clinical modifier", "Mode of inheritance", "Past medical history", "Blood group", and "Frequency"—all very useful for describing phenotypes from an ontological perspective, but harboring the potential to create uninformative connections that may affect topology reliant clustering methods. For example, the HPO term "Autosomal recessive inheritance" has 2,762 direct neighbors, only two of which are HPO terms. By including this hub, the genes *MPIG6B* and *RIPOR2* are connected by the path *MPIG6B*—"Autosomal recessive inheritance"—*RIPOR2*, whereas without the nodes encoding modes of inheritance, the shortest paths connecting *MPIG6B* to *RIPOR2* are of length 6. Pruning and curating just a single ontology did not present much of a challenge to our study, but with the vast number of large and complex ontologies available, if someone were to try and include a large quantity of them in their own graph could present serious undertaking in selecting which parts of an ontology to include to avoid potential spurious connections.

In this study, we devised a snowball-sampling-based null model and showed it to be more conservative than null models used in other works based on the generation of random clusters and edge-shuffling that have been used previously to quantify cluster and model significance [23]. By combining this null model with the time series aspect of our network (Section), we have created a robust null model and evaluation method. This snowballing null model and edge-rediscovery is another contribution that could easily be taken and applied broadly in the evaluation of future methods and analyses.

The entire application of clustering of biological networks to study disease rests on the idea of guilt-by-association—which means genes that are functionally related or co-expressed with a disease-associated gene or protein are also likely to be involved in the same disease or biological process. Guilt-by-association is an assumption and is known to be imperfect [11]. BOCC uses and compares a variety of widely used clustering algorithms. In our comparisons, we concluded that the best results are achieved as an ensemble method rather than any single one.

Additionally, through our analyses, we found that the vast majority of identified clusters do not exceed the expectations set by our null model; in fact only 16.8% of non-trivial clusters do. These clusters are predicted to be biologically relevant and useful for hypothesis generation but may or may not identify genetic pathways [17–19] or disease modules [12–16]—previous research has shown similar clustering algorithms identify these types of information but we do not explicitly measure it. Using a variety of network topology and biological clustering measurements we were able to train a classification model to predict the significance of a cluster with a reasonable degree of certainty by identifying those clusters where guilt-by-association is valid.

BOCC relies heavily on the structure of the PPI networks and phenotype ontology that make up its knowledge graph. While these sources provide much of the power behind BOCC, they also bring with them inherent biases. The growth of this source is not consistent or unbiased. STRING is an aggregation of numerous databases, PPI networks, and studies. It has been shown that the growth and structure of PPI networks is not necessarily representative of biology, but of the interest and focus of the biomedical research field as a whole [50]. Similarly, the depth and breadth of phenotype terms vary widely throughout HPO. For instance the term "Arthrogryposis multiplex congenita", HP:0002804 is a leaf node in HPO, the most specific available term, but is associated with numerous very specific and distinct types of arthrogryposis in OMIM such as "Arthrogryposis, distal, type 1A" and "Arthrogryposis, distal, type 2B2". This is an area within HPO that could potentially benefit from deeper phenotyping. But it also highlights the power of the hierarchical nature of the ontology as patients with any type of arthrogryposis can still be described with a more general but still fairly specific HPO term. Regardless of the limitations of STRING and HPO, they remain best in class for what they are.

In our experiment using the patient profiles, we consider patients "undiagnosed" if there is no direct connection between the variant harboring genes and HPO terms. This is a simplification as a patient's diagnostic state is not determined by any computation tool, ontology, or database. These resources aid decisions but diagnoses are ultimately made by health care professionals and their careful consideration of the evidence. Our assumption is however necessary and useful if useful for trialing our tool.

In this study we operated on a relatively small knowledge graph, using only two source ontologies. In Open Biological and Biomedical Ontology Foundry (OBO Foundry) [51] there are well over 100 ontologies currently maintained and active. There are many ongoing efforts to combine all knowledge relating to human health into a single large knowledge graph such as PheKnowlater [52] and the Integrated Monarch Ontology [53]. A clear future direction of this work is to include, learn from, and use a more full representation of human biology. One hurdle to overcome here is the semantic information also accompanying biological ontologies. Just as we had to prune HPO prior to using it in this study, we anticipate the same would be required for other sources, which may become a burden as the number of ontologies grows. A method like BOCC could also be expanded by integrating quantifiable data like gene expression. Work already exists using this data for phenotype classification in depended of biological networks or ontologies [54].

In scaling up the quantity of information used, we anticipate other algorithmic approaches may become necessary. One of the great advantages of using a full knowledge graph is the vast heterogeneity of the data types. Instead of operating on just genes and phenotypes, we could include diseases, drug interactions, pathways, regulator elements, tissue expression/specificity, and much more. However, most clustering algorithms are blind to node and edge type, a huge loss of information. Of the clustering algorithms we used here, only CESNA took into account node type. This problem inspired the recent ECoHeN algorithm [55] and is also addressed in proposed alterations to modularity to account for heterogeneous networks [56]; though this

latter approach may still be hindered by the resolution limit inherent to modularity-based approaches [57]. Alternatively, it has been shown how node embeddings can be created and used to represent higher-order patterns in biological knowledge graphs more heterogeneous than those used here [16].

Through this study, we have demonstrated how a variety of network science clustering methods applied to a heterogeneous biological time-varying network can be instrumental in finding novel connections between mutation-harboring genes and phenotypes, potentially causal links for rare and undiagnosed diseases. From a conceptual standpoint, this study contributes to the methodological understanding of clustering and classifying in network biology for understanding rare and undiagnosed diseases. It also provides concrete and ready-to-use tools for exploring the gene-to-phenotype connections in our clusters in a web application and command line interface.

## Supporting information

**S1 Fig. A contrived example illustrating various types of community structure including a) disjoint, B) overlapping, and C) hierarchical communities.** Adapted from [58].
(TIFF)

**S2 Fig.** Shown are distributions of cluster size using A. greedy B. walktrap C. infomap and D. censa. The second column contains the subcluster size distributions, where every cluster from panels A-D were clustered again with the paris-hierarchical method E. greedy-paris F. walktrap-paris G. infomap-paris and H. censa-paris. In all of these plots the x-axis is cluster-size and the y-axis is the number of clusters. The first three of the clustering algorithms on their own, have a tendency to produce very few clusters that are all very large, some with a membership larger than 20,000 nodes, in the case of infomap. Our end goal is to use these clusters to provide sets of genes and phenotypes that are likely to have yet-to-be-discovered clinically meaningful relationships. These clusters in A-D are far too large to be useful for hypothesis generation in clinical or experimental settings. A second layer of cluster with the paris method is applied and shown in F-H, setting an upper limit of 100 on cluster size results in many more clusters of a size manageable for human curation.
(TIFF)

**S3 Fig. Distribution of empirical p-values of the 2019 greedy-paris clusters using three different null models: Snowballing sampling, edge shuffle, and random clusters.** All figures share the same y-axis which is also on a log scale. The proportion of clusters with $p < 0.05$ in each model is 6%, 48%, and 39%, for snowballing, edge-shuffle, and random clusters respectively. Snowballing has 70% of its clusters with $p = 1.00$, whereas the other two models have zero clusters falling into this category.
(PDF)

**S1 Methods. Additional details about clustering methods and alternative null models.**
(PDF)

**S1 Dataset. Data from MyGene2.**
(CSV)

## Acknowledgments

The authors thank Dr. Jessica Chong for her help in understanding MyGene2, and Dr. Tiffany Callahan for her helpful feedback and advice along the way.

## Author Contributions

**Conceptualization:** Michael S. Bradshaw, Bailey Fosdick, Ryan Layer.

**Data curation:** Michael S. Bradshaw, Taylor Firman, Alisa Gaskell.

**Formal analysis:** Michael S. Bradshaw.

**Funding acquisition:** Ryan Layer.

**Investigation:** Connor Gibbs.

**Methodology:** Michael S. Bradshaw, Connor Gibbs.

**Resources:** Taylor Firman, Alisa Gaskell.

**Software:** Michael S. Bradshaw, Skylar Martin.

**Supervision:** Alisa Gaskell, Bailey Fosdick, Ryan Layer.

**Visualization:** Michael S. Bradshaw.

**Writing – original draft:** Michael S. Bradshaw.

**Writing – review & editing:** Michael S. Bradshaw, Connor Gibbs, Bailey Fosdick, Ryan Layer.

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
