## [Decision Letter · Decision Letter 0]

8 Jan 2024

PONE-D-23-41162HYPOTHESIS GENERATION FOR RARE AND UNDIAGNOSED DISEASES THROUGH CLUSTERING AND CLASSIFYING TIME-VERSIONED BIOLOGICAL ONTOLOGIESPLOS ONE

Dear Dr. Layer,

Thank you for submitting your manuscript to PLOS ONE. After careful consideration, we feel that it has merit but does not fully meet PLOS ONE’s publication criteria as it currently stands. Therefore, we invite you to submit a revised version of the manuscript that addresses the points raised during the review process.

**ACADEMIC EDITOR: **

The main comments made by the reviewers on the manuscript about the Biological Ontology Cluster Classification (BOCC) tool can be summarized as follows:

1. Figures in the manuscript are not properly labeled.

2. Inconsistency in font face and clarity between text and images.

3. Lack of clarity on the novelty of the study needs highlighting in the introduction.

4. Absence of a literature review section to establish context and credibility.

5. The manuscript seems outdated due to the limited citation of recent works; it needs updating with suggested references made by the reviewers, which I suggest you incorporate in your article, such as the introduction section or literature review section.

6. Recommendations for proofreading to correct grammatical mistakes.

7. Suggestion to compare results with state-of-the-art studies for validation.

8. Absence of a discussion on the limitations of the study; recommended for inclusion in the conclusion.

9. Overall, the paper is well-written and suitable for the journal but lacks a conclusion section to summarize findings and implications.

This summary addresses the key areas the reviewers have identified for improvement, including technical presentation, content depth, literature context, and overall structure.

We look forward to receiving your revised manuscript.

Kind regards,

Roseline Oluwaseun Ogundokun, Ph.D.

Academic Editor

PLOS ONE

Journal Requirements:

2. Please note that PLOS ONE has specific guidelines on code sharing for submissions in which author-generated code underpins the findings in the manuscript. In these cases, all author-generated code must be made available without restrictions upon publication of the work. Please review our guidelines at https://journals.plos.org/plosone/s/materials-and-software-sharing#loc-sharing-code and ensure that your code is shared in a way that follows best practice and facilitates reproducibility and reuse

3. Please note that PLOS ONE has specific guidelines on code sharing for submissions in which author-generated code underpins the findings in the manuscript. In these cases, all author-generated code must be made available without restrictions upon publication of the work. Please review our guidelines at https://journals.plos.org/plosone/s/materials-and-software-sharing#loc-sharing-code and ensure that your code is shared in a way that follows best practice and facilitates reproducibility and reuse

4. We suggest you thoroughly copyedit your manuscript for language usage, spelling, and grammar. If you do not know anyone who can help you do this, you may wish to consider employing a professional scientific editing service.  

5. Thank you for stating the following financial disclosure: "This work was supported by a grant from Children's Hospital Colorado."  

6. Thank you for stating the following in the Acknowledgments Section of your manuscript: "This work was supported by a grant from Children's Hospital Colorado."  

Please remove any funding-related text from the manuscript and let us know how you would like to update your Funding Statement. Currently, your Funding Statement reads as follows: "This work was supported by a grant from Children's Hospital Colorado."  

7. We note that you have indicated that there are restrictions to data sharing for this study. For studies involving human research participant data or other sensitive data, we encourage authors to share de-identified or anonymized data. However, when data cannot be publicly shared for ethical reasons, we allow authors to make their data sets available upon request. For information on unacceptable data access restrictions, please see http://journals.plos.org/plosone/s/data-availability#loc-unacceptable-data-access-restrictions. 

9. We notice that your supplementary figures are uploaded with the file type 'Figure'. Please amend the file type to 'Supporting Information'. Please ensure that each Supporting Information file has a legend listed in the manuscript after the references list.

Additional Editor Comments:

The main comments made by the reviewers on the manuscript about the Biological Ontology Cluster Classification (BOCC) tool can be summarized as follows:

1. Figures in the manuscript are not properly labeled.

2. Inconsistency in font face and clarity between text and images.

3. Lack of clarity on the novelty of the study needs highlighting in the introduction.

4. Absence of a literature review section to establish context and credibility.

5. The manuscript seems outdated due to the limited citation of recent works; it needs updating with suggested references made by the reviewers, which I suggest you incorporate in your article, such as the introduction section or literature review section.

6. Recommendations for proofreading to correct grammatical mistakes.

7. Suggestion to compare results with state-of-the-art studies for validation.

8. Absence of a discussion on the limitations of the study; recommended for inclusion in the conclusion.

9. Overall, the paper is well-written and suitable for the journal but lacks a conclusion section to summarize findings and implications.

This summary addresses the key areas the reviewers have identified for improvement, including technical presentation, content depth, literature context, and overall structure.

Reviewers' comments:

Reviewer's Responses to Questions

**Comments to the Author**

1. Is the manuscript technically sound, and do the data support the conclusions?

Reviewer #1: Yes

Reviewer #2: Yes

Reviewer #3: Yes

2. Has the statistical analysis been performed appropriately and rigorously? 

Reviewer #1: Yes

Reviewer #2: Yes

Reviewer #3: Yes

3. Have the authors made all data underlying the findings in their manuscript fully available?

Reviewer #1: Yes

Reviewer #2: Yes

Reviewer #3: Yes

4. Is the manuscript presented in an intelligible fashion and written in standard English?

Reviewer #1: No

Reviewer #2: Yes

Reviewer #3: Yes

5. Review Comments to the Author

Reviewer #1: the study presents an innovative tool named Biological Ontology Cluster Classification (BOCC). This tool is aimed at assisting in the diagnosis of rare and undiagnosed diseases by identifying potential gene-to-phenotype (g2p) associations that are not explicitly documented in the current literature. Here are my comments on the manuscript:

1. All the figures are not properly labeled.

2. the manuscript has different font face. for instance, I dont know how I will classify the text/image between line 50 and 51. Is it text or image, if it is text, while did it have different front face and if it is image while is it not label?

3. I understand that BOCC is available as both a web application and a command-line tool, making it accessible for different types of users ranging from researchers to clinicians. However, what is the novelty of this study? the author is expected to highlights the contributions of this study towards the last paragraph of the introduction section.

4. I suggest that the author should create a section to discuss the literature review of some related work done in this area so as to enhance the credibility of this study.

6. Havn't checked this manuscript thoroughly, I observed that the author makes little efforts to cite and reference 2023 work. This act makes the manuscript looks outdated. Therefore, I will suggest that the author should make use of this following searched references to updated their manuscript:

1. Zhuang, Y., Jiang, N., Xu, Y., Xiangjie, K., & Kong, X. (2022). Progressive Distributed and Parallel

Similarity Retrieval of Large CT Image Sequences in Mobile Telemedicine Networks. Wireless

communications and mobile computing, 2022. doi: 10.1155/2022/6458350

2. Lu, G., Duan, L., Meng, S., Cai, P., Ding, S.,... Wang, X. (2023). Development of a colorimetric and

turn-on fluorescent probe with large Stokes shift for H2S detection and its multiple applications

in environmental, food analysis and biological imaging. Dyes and Pigments, 220, 111687. doi:

https://doi.org/10.1016/j.dyepig.2023.111687

3. Siyu Lu, J. Y. B. Y. (2023). Analysis and Design of Surgical Instrument Localization Algorithm.

Computer Modeling in Engineering & Sciences, 137(1), 669-685. doi:

10.32604/cmes.2023.027417

4. Zhu, Y., Huang, R., Wu, Z., Song, S., Cheng, L.,... Zhu, R. (2021). Deep learning-based predictive

identification of neural stem cell differentiation. Nature Communications, 12(1), 2614. doi:

10.1038/s41467-021-22758-0

5. Chen, L., He, Y., Zhu, J., Zhao, S., Qi, S., Chen, X.,... Xie, T. (2023). The roles and mechanism of

m6A RNA methylation regulators in cancer immunity. Biomedicine & Pharmacotherapy, 163,

114839. doi: https://doi.org/10.1016/j.biopha.2023.114839

6. Huang, H., Liu, L., Wang, J., Zhou, Y., Hu, H., Ye, X.,... Tang, B. Z. (2022). Aggregation caused

quenching to aggregation induced emission transformation: a precise tuning based on BN-doped

polycyclic aromatic hydrocarbons toward subcellular organelle specific imaging. Chemical

Science, 13(11), 3129-3139. doi: 10.1039/D2SC00380E

7. Huang, H., Wu, N., Liang, Y., Peng, X., & Shu, J. (2022). SLNL: A novel method for gene selection

and phenotype classification. International Journal of Intelligent Systems, 37(9), 6283-6304. doi:

https://doi.org/10.1002/int.22844

5. I suggest that the manuscript should be thoroughly proof-read to avoid some grammar mistake.

6. I recommend that the author should compare their result with the state-of-the-art studies so as to validate the strength of the obtained result.

7. What is the limitation of this study. This can be included in the conclusion section to give room for future research.

Reviewer #2: This manuscript is well written, introducing their BOCC tool. As they write, BOCC is a series of network-science-based methodologies that identify relevant clusters from a heterogeneous network comprised of HPO, STRING, OMIM, and Orphanet. I do not have any major suggestions for how to improve this paper, and it seems suitable for this journal venue.

Reviewer #3: The research addressed the topic of discuss but there are areas to be improved. The importance of literature review cannot be overemphasized. it is of great importance as it serves the purpose of establishing the context of a research study by elucidating the existing knowledge on the subject matter. This aids researchers in situating their work within the already established body of knowledge. Secondly, it allows researchers to identify gaps within the current knowledge base, which can subsequently form the foundation for research questions or hypotheses. Thirdly, a literature review supports the justification of research methodologies or methods by showcasing the manner in which prior studies were conducted and their contributions to the field. Therefore as observed that this important section is missing in the work, I suggest that it be included and relevant work be added to improve this research. I suggest the following work:

1. Luo, Y., Chen, D., & Xing, X. (2023). Comprehensive Analyses Revealed Eight Immune Related Signatures Correlated With Aberrant Methylations as Prognosis and Diagnosis Biomarkers for Kidney Renal Papillary Cell Carcinoma. Clinical Genitourinary Cancer, 21(5), 537-545. doi: https://doi.org/10.1016/j.clgc.2023.06.011

2. Gan, Y., Xu, Y., Zhang, X., Hu, H., Xiao, W., Yu, Z.,... Zheng, S. (2023). Revisiting Supersaturation of a Biopharmaceutical Classification System IIB Drug: Evaluation via a Multi-Cup Dissolution Approach and Molecular Dynamic Simulation. Molecules , 28(19), 6962. doi: https://doi.org/10.3390/molecules28196962

3. Fan, Z., He, Y., Sun, W., Li, Z., Ye, C.,... Wang, C. (2023). Clinical characteristics, diagnosis and management of Sweet syndrome induced by azathioprine. Clinical and Experimental Medicine, 23, 3581-3587. doi: 10.1007/s10238-023-01135-9

4. Wu, J., Fang, Z., Wang, X., Zeng, W., Zhao, Y., Jiang, F.,... Li, J. (2022). SLIT2 Rare Sequencing Variants Identified in Idiopathic Hypogonadotropic Hypogonadism. Hormone Research in Paediatrics, 95(4), 384-392. doi: 10.1159/000525769

5. Gong, T., Zhang, F., Feng, L., Zhu, X., Deng, D., Ran, T.,... Ji, X. (2023). Diagnosis and surgical outcomes of coarctation of the aorta in pediatric patients: a retrospective study. Frontiers in Cardiovascular Medicine, 10. doi: 10.3389/fcvm.2023.1078038

6. Fan, Z., He, Y., Sun, W., Li, Z., Ye, C., & Wang, C. (2023). Amoxicillin-induced aseptic meningitis: clinical features, diagnosis and management. European journal of medical research, 28(1), 301. https://doi.org/10.1186/s40001-023-01251-y

7. Jin, K., Gao, Z., Jiang, X., Wang, Y., Ma, X., Li, Y.,... Ye, J. (2023). MSHF: A Multi-Source Heterogeneous Fundus (MSHF) Dataset for Image Quality Assessment. Scientific Data, 10(1), 286. doi: 10.1038/s41597-023-02188-x

The introduction should include wht the work is contribution to the body of knowledge and conclude with section of the work.

The conclusion furnishes a brief overview of the primary discoveries and outcomes of the investigation, highlighting the significant implications of the study. It permits the researcher to contemplate on the degree to which the study has accomplished its objectives and whether the research inquiries have been addressed. Implications: The conclusion provides an occasion to deliberate upon the ramifications of the research findings, encompassing their pertinence to the research domain, practical implementations, and prospective influence.

I observed that there is no conclusion. I suggest this section be included.

6. PLOS authors have the option to publish the peer review history of their article (what does this mean?). If published, this will include your full peer review and any attached files.

Reviewer #1: No

Reviewer #2: No

Reviewer #3: No

---

## [Author Response · Author response to Decision Letter 0]

20 Feb 2024

Rebuttal Letter

The text in black is copied and pasted from the email from the editor. Our comments and

response to each item is in red.

1. Please ensure that your manuscript meets PLOS ONE's style requirements, including those for file naming. The PLOS ONE style templates can be found at https://journals.plos.org/plosone/s/file?id=wjVg/PLOSOne_formatting_sample_main_body.pdf and https://journals.plos.org/plosone/s/file?id=ba62/PLOSOne_formatting_sample_title_authors_affili ations.pdf

Updated formatting

2. Please note that PLOS ONE has specific guidelines on code sharing for submissions in which author-generated code underpins the findings in the manuscript. In these cases, all author-generated code must be made available without restrictions upon publication of the work. Please review our guidelines at https://journals.plos.org/plosone/s/materials-and-software-sharing#loc-sharing-code and ensure that your code is shared in a way that follows best practice and facilitates reproducibility and reuse

License added to github, Data Availability Statement updated

3. Please note that PLOS ONE has specific guidelines on code sharing for submissions in which author-generated code underpins the findings in the manuscript. In these cases, all author-generated code must be made available without restrictions upon publication of the work. Please review our guidelines at https://journals.plos.org/plosone/s/materials-and-software-sharing#loc-sharing-code and ensure that your code is shared in a way that follows best practice and facilitates reproducibility and reuse

Data Availability Statement section has been added with details about code and data access.

4. We suggest you thoroughly copyedit your manuscript for language usage, spelling, and grammar. If you do not know anyone who can help you do this, you may wish to consider employing a professional scientific editing service.

 Whilst you may use any professional scientific editing service of your choice, PLOS has partnered with both American Journal Experts (AJE) and Editage to provide discounted services to PLOS authors. Both organizations have experience helping authors meet PLOS guidelines and can provide language editing, translation, manuscript formatting, and figure formatting to ensure your manuscript meets our submission guidelines. To take advantage of our partnership with AJE, visit the AJE website (http://learn.aje.com/plos/) for a 15% discount off AJE services. To take advantage of our partnership with Editage, visit the Editage website (http://www.editage.com ) and enter referral code PLOSEDIT for a 15% discount off Editage services. If the PLOS editorial team finds any language issues in text that either AJE or Editage has edited, the service provider will re-edit the text for free.

Edits were made by Michael Bradshaw and Ryan Layer. Changes were tracked with Overleaf and will be provided.

5. Thank you for stating the following financial disclosure: "This work was supported by a grant from Children's Hospital Colorado."

Elaborated members of Children’s Hospital were involved in this study saying:

This work was supported by a grant from Children's Hospital Colorado. Members of the funding body collected the patient data, aided in the direction of analysis, and are authors of the paper.

6. Thank you for stating the following in the Acknowledgments Section of your manuscript: "This work was supported by a grant from Children's Hospital Colorado."

Please remove any funding-related text from the manuscript and let us know how you would like to update your Funding Statement. Currently, your Funding Statement reads as follows: "This work was supported by a grant from Children's Hospital Colorado."

Amended text has been added to the cover letter.

7. We note that you have indicated that there are restrictions to data sharing for this study. For studies involving human research participant data or other sensitive data, we encourage authors to share de-identified or anonymized data. However, when data cannot be publicly shared for ethical reasons, we allow authors to make their data sets available upon request. For information on unacceptable data access restrictions, please see http://journals.plos.org/plosone/s/data-availability#loc-unacceptable-data-access-restrictions.

Data Availability statement has been updated to include details about requesting access to CHCO patient records. The patient data from MyGene2, all of which is publicly available is included in S1_dataset.csv

Supporting info section added

 9. We notice that your supplementary figures are uploaded with the file type 'Figure'. Please amend the file type to 'Supporting Information'. Please ensure that each Supporting Information file has a legend listed in the manuscript after the references list.

Supplementary figures have been properly labeled

Duplicate citations removed

Additional Editor Comments:

The main comments made by the reviewers on the manuscript about the Biological Ontology Cluster Classification (BOCC) tool can be summarized as follows:

1. Figures in the manuscript are not properly labeled.

Figures labels have been updated to match PLOS guidelines.

2. Inconsistency in font face and clarity between text and images.

3. Lack of clarity on the novelty of the study needs highlighting in the introduction. 4. Absence of a literature review section to establish context and credibility.

A literature review has been added to the introduction.

5. The manuscript seems outdated due to the limited citation of recent works; it needs updating with suggested references made by the reviewers, which I suggest you incorporate in your article, such as the introduction section or literature review section.

Additional references have been added in the literature review section.

6. Recommendations for proofreading to correct grammatical mistakes.

Proof reading has been done and edits made

7. Suggestion to compare results with state-of-the-art studies for validation.

We agree with you about the importance of comparing tools. Our manuscript includes comparisons with widely used clustering algorithms and null models. More specifically, we

 compared four pre-existing and widely used clustering algorithms and a comparison of the traditional random-graph based null models vs the more conservative snow-ball sampling-based approach. To make this easier to find, we have added clearer language about our comparisons and findings. As for comparing high level results of BOCC to other tools, we do not know of any other method that unbiasedly discovers co-occurring genotype/phenotype pairs. While it is module identification tools and variant prioritization tools do yield pairs, their search is biases toward pathways or patients. If the reviewers are aware of tool directly comparable to BOCC that we are not, we will be happy to perform a comparison and cite them.

8. Absence of a discussion on the limitations of the study; recommended for inclusion in the conclusion.

We agree on the importance of a discussion of the limitations, discussion has been added.

9. Overall, the paper is well-written and suitable for the journal but lacks a conclusion section to summarize findings and implications.

Per PLOS ONE submission and formatting guidelines a conclusion section is not required and can be combined with the discussion and or results section. See https://journals.plos.org/plosone/s/submission-guidelines#loc-results-discussion-conclusions

“Results, Discussion, Conclusions

These sections may all be separate, or may be combined to create a mixed Results/Discussion section (commonly labeled “Results and Discussion”) or a mixed Discussion/Conclusions section (commonly labeled “Discussion”).”

We found for the sake of narrative integrating the discussion and conclusion seemed best. But we acknowledge that if the reviewers miss that information it should be clearer, so we have added emphasising language in the discussion section about we conclude.

This summary addresses the key areas the reviewers have identified for improvement, including technical presentation, content depth, literature context, and overall structure.

Reviewers' comments:

Reviewer's Responses to Questions Comments to the Author

1. Is the manuscript technically sound, and do the data support the conclusions?

Reviewer #1: Yes

Reviewer #2: Yes

Reviewer #3: Yes

2. Has the statistical analysis been performed appropriately and rigorously? Reviewer #1: Yes

Reviewer #2: Yes

Reviewer #3: Yes

3. Have the authors made all data underlying the findings in their manuscript fully available?

Reviewer #1: Yes

Reviewer #2: Yes

Reviewer #3: Yes

4. Is the manuscript presented in an intelligible fashion and written in standard English?

Reviewer #1: No Reviewer #2: Yes

 Reviewer #3: Yes

We and collegaes have done a general full proofreading of the manuscript to catch any errors.

5. Review Comments to the Author

Reviewer #1: the study presents an innovative tool named Biological Ontology Cluster Classification (BOCC). This tool is aimed at assisting in the diagnosis of rare and undiagnosed diseases by identifying potential gene-to-phenotype (g2p) associations that are not explicitly documented in the current literature. Here are my comments on the manuscript:

1. All the figures are not properly labeled.

Thank you for noting this, figure labels have been fixed.

2. the manuscript has different font face. for instance, I dont know how I will classify the text/image between line 50 and 51. Is it text or image, if it is text, while did it have different front face and if it is image while is it not label?

Formatting of this section has been edited to match PLOS’s guidelines.

3. I understand that BOCC is available as both a web application and a command-line tool, making it accessible for different types of users ranging from researchers to clinicians. However, what is the novelty of this study? the author is expected to highlights the contributions of this study towards the last paragraph of the introduction section.

The novelty of BOCC has now been highlighted in the introduction, it was previously only discussed at length in the Discussion section.

4. I suggest that the author should create a section to discuss the literature review of some related work done in this area so as to enhance the credibility of this study.

A literature review has been added to the introduction, complete with a table!

6. Havn't checked this manuscript thoroughly, I observed that the author makes little efforts to cite and reference 2023 work. This act makes the manuscript looks outdated. Therefore, I will suggest that the author should make use of this following searched references to updated their manuscript:

 Thank you for this suggestion to include more recent citations. We have included additional references in the literature review and discussion sections. We have carefully reviewed each paper listed below and comment on why we chose to cite it or not.

1. Zhuang, Y., Jiang, N., Xu, Y., Xiangjie, K., & Kong, X. (2022). Progressive Distributed and Parallel

Similarity Retrieval of Large CT Image Sequences in Mobile Telemedicine Networks. Wireless communications and mobile computing, 2022. doi: 10.1155/2022/6458350

We appreciate this article deals with human health and 

---

## [Decision Letter · Decision Letter 1]

15 May 2024

PONE-D-23-41162R1HYPOTHESIS GENERATION FOR RARE AND UNDIAGNOSED DISEASES THROUGH CLUSTERING AND CLASSIFYING TIME-VERSIONED BIOLOGICAL ONTOLOGIESPLOS ONE

Dear Dr. Layer,

Thank you for submitting your manuscript to PLOS ONE. After careful consideration, we feel that it has merit but does not fully meet PLOS ONE’s publication criteria as it currently stands. Therefore, we invite you to submit a revised version of the manuscript that addresses the points raised during the review process.

**I would like to sincerely apologise for the delay you have incurred with your submission. Due to the concerns raised about the process of evaluation of this manuscript, it was considered necessary to invite additional reviewers to provide comments on your study. Although some reviewers are happy with the revised version, other reviewers have raised remaining scientific concerns about the study that need to be addressed. Please revise the manuscript to address all the reviewer's comments in a point-by-point response in order to ensure it is meeting the journal's publication criteria. Please note that the revised manuscript will need to undergo further review, we thus cannot at this point anticipate the outcome of the evaluation process.**

We look forward to receiving your revised manuscript.

Kind regards,

Miquel Vall-llosera Camps

Staff Editor

PLOS ONE

Reviewers' comments:

Reviewer's Responses to Questions

**Comments to the Author**

1. If the authors have adequately addressed your comments raised in a previous round of review and you feel that this manuscript is now acceptable for publication, you may indicate that here to bypass the “Comments to the Author” section, enter your conflict of interest statement in the “Confidential to Editor” section, and submit your "Accept" recommendation.

Reviewer #1: All comments have been addressed

Reviewer #3: All comments have been addressed

Reviewer #4: All comments have been addressed

Reviewer #5: (No Response)

2. Is the manuscript technically sound, and do the data support the conclusions?

Reviewer #1: Yes

Reviewer #3: Yes

Reviewer #4: Yes

Reviewer #5: Partly

3. Has the statistical analysis been performed appropriately and rigorously? 

Reviewer #1: Yes

Reviewer #3: Yes

Reviewer #4: Yes

Reviewer #5: Yes

4. Have the authors made all data underlying the findings in their manuscript fully available?

Reviewer #1: Yes

Reviewer #3: Yes

Reviewer #4: Yes

Reviewer #5: Yes

5. Is the manuscript presented in an intelligible fashion and written in standard English?

Reviewer #1: Yes

Reviewer #3: Yes

Reviewer #4: Yes

Reviewer #5: No

6. Review Comments to the Author

**Reviewer #1**: The Author have attended to all the comments raised by the reviewer. Therefore, I recommend the manuscript for publication.

**Reviewer #3**: (No Response)

**Reviewer #4:** The study is novel and Article may be accepted. This research is fascinating and incredibly promising for advancing our understanding of rare diseases and providing potential diagnoses for patients who have struggled to find answers. The approach of integrating protein-protein interactions and phenotype relationships through networks and clustering is innovative and holds great potential for uncovering latent gene-to-phenotype connections. The use of databases like STRING and HPO, combined with the development of a tool like BOCC, marks a significant step forward in addressing the diagnostic challenges posed by rare diseases. The ability to identify significant clusters that correlate with known drug interactions is particularly exciting, as it opens new avenues for targeted therapeutic interventions.

**Reviewer #5:** The authors introduced a clustering method that can be used to discover the potential and novel gene-phenotype relationship. It is especially important for undiagnosed patients and unknown gene-phenotype relationships. The clusters reported by the method can be a guide to shortening the search space. Although the author addressed many questions raised by the reviewers in the first review process, the experimental setup and the clinical usage and validation are still unclear to me. Moreover, the author needs to do more proofreading as numerous mistakes are observed easily.

Major comments:

1. The formatting and language are usually relatively minor during review. However, multiple reviewers have already pointed it out in the first review. Several grammatical errors and formatting errors persist in the manuscript. I think the author should take it very seriously. I still observed many formatting issues, such as a missing space before the citation. For example, in page 2, “Matchmaker Exchange[2] and MyGene2[1]”. Many issues might be easily detected by Grammarly or other tools. I will recommend really finding an English editing service for the proofreading.

2. In page 2, these two sentences: “Using the Human Phenotype Ontology, we can expand the patient’s assigned phenotypes to include all closely related phenotypes and diseases. By connecting STRING and HPO [4] with the gene-to-phenotype (g2p) connections from Orphanet[5] and OMIM[6], we can then look for indirect associations between the patient’s assigned data points.”

HPO was not introduce as abbriveation after the first appearance. And the citation for HPO should be after the first appearance.

3. Page 3: “There is a trend toward using an ensemble of methods or methods capable of using higher-order patterns and integrating multiple types of networks (Table 1 ).” Redundant space before “)”.

4. Page 10: “Fig 4. A. Procedure for training and evaluating the XGboost model. Features were generated about each cluster (identified as described in Section )”, which Section? Is the Section number missing here? In page 11: the same issue, “After performing quality control (Section ), there were on average 401 variants per patient (Fig 6.C).”

5. The author refers g2p to gene to phenotype. Does the author use Phenotype in this paper as a feature (HPO term)? We often use phenotype to refer to disorders and describe HPO terms by features.

6. The author's definition of the undiagnosed patient in CHCO and MyGene2 is unclear. The first sentence of the result section states, “We applied BOCC to 721 patients from Children’s Hospital Colorado (CHCO) with suspected genetic drivers of their diseases.” Does that mean these patients are undiagnosed?

7. Moreover, in MyGene2 section, “We found that presently 111 of these profiles contained no direct connection between any of their genes and any of their HPO terms; we assumed these cases to still be undiagnosed.”, I think it is problematic. It is dangerous to assume the patient is undiagnosed because there is no connection between HPO and the gene. I randomly checked some patients in Table S1 and Table 5. I found some of the patients are already published in the literature, and I believe they are diagnosed. The first question is the patient with ID 3071 (https://mygene2.org/MyGene2/familyprofile/3071/profile). It is the first patient in Table S1. This patient was already published in a paper with PMID (25142838) and annotated with Kabuki and KMT2D. Does the author consider this patient as diagnosed or undiagnosed? For example, patient ID 877 (https://mygene2.org/MyGene2/familyprofile/877/genetic/gene) is already published in https://www.ncbi.nlm.nih.gov/pmc/articles/PMC5082428/. This patient was also annotated with Joubert syndrome and a likely pathogenic mutation in B9D2. Besides, patient ID 1292 has likely pathogenic mutation in TNNT3 and was diagnosed as “Arthrogryposis, Distal, Type 1A”. And the author reported “Arthrogryposis multiplex congenita” as a new relationship to TNNT3. However, we can see from the disease name linked to TNNT3, which is “Arthrogryposis, distal, type 2B2,” and the entry in omim was updated in 2019. I would say this new relationship between this HPO and TNNT3 is not surprised at all because you can see it from the disease name. Therefore, I wonder what the meaning of this prediction is.

8. My main concern is that I don’t know how I validate wheter the connection found by the cluster is correct or not. The experiement should start from validate on the existing relationship from diagnosed patients. Then we know that the relationship found in the cluster do have some meanings. Then, we might select some patients with a disease gene found after the model is built. For example, the model trained on the data from 2019, then we can check whether the model can predict the relationship between the HPO and a Gene X, which was found as a disease-causing gene in 2020. As pointed out previously, when the definition of diagnosed and undiagnosed is unclear, it is very difficult to validate whether the cluster is meaningful or not, even though the experiments reported many significant p-values from different methods.

9. In the end, how the clinicians use this tool is unclear. There are many nodes in the graph, and many clusters were reported after the HPO and genes are given as input. It will be great to show more examples and the steps how the user should use. For example, I tried several HPOs linked to Cornelia de Lange syndrome (NIPBL gene), Long eyelashes (HP:0000527). Synophrys (HP:0000664) and Highly arched eyebrow (HP:0002553). However, I only found NIPBL is the third claster. I believe it is the key features for CdLS (NIPBL). Therefore, how do I interpret the first two clusters without NIPBL? The experiment and results showed that this method is working, statistically. However, how to interpret the results and validate the clinical meaning is unclear.

10. Will the gene in the same pathway or same phenotypic series shown in the same cluster?

11. In author’s first example, Patient 1930 with VUS in NBEA and SSPO. The author stated in the introduction that with more experiment to the associated phenotype in the clusters could help us to diagnose this patient. I believe the phenotype here is the HPO term. I wonder why we can use the new gene-to-hpo to solve the case with VUS? Does it contribute to the ACMG variant classification?

7. PLOS authors have the option to publish the peer review history of their article (what does this mean?). If published, this will include your full peer review and any attached files.

Reviewer #1: No

Reviewer #3: No

Reviewer #4: **Yes: **Amit Joshi

Reviewer #5: **Yes: **Tzung-Chien Hsieh

---

## [Author Response · Author response to Decision Letter 1]

25 Jun 2024

Response to Reviewers

Major comments:

1. The formatting and language are usually relatively minor during review. However, multiple reviewers have already pointed it out in the first review. Several grammatical errors and formatting errors persist in the manuscript. I think the author should take it very seriously. I still observed many formatting issues, such as a missing space before the citation. For example, in page 2, “Matchmaker Exchange[2] and MyGene2[1]”. Many issues might be easily detected by Grammarly or other tools. I will recommend really finding an English editing service for the proofreading.

We apologize for the mistakes, thank you for bringing them to our attention. We have used grammarly and have had multiple native English speakers proofread and edit the manuscript for this revision. We hope you find it satisfactory.

2. In page 2, these two sentences: “Using the Human Phenotype Ontology, we can expand the patient’s assigned phenotypes to include all closely related phenotypes and diseases. By connecting STRING and HPO [4] with the gene-to-phenotype (g2p) connections from Orphanet[5] and OMIM[6], we can then look for indirect associations between the patient’s assigned data points.”

HPO was not introduce as abbriveation after the first appearance. And the citation for HPO should be after the first appearance.

Thank you for your attention to detail; we have fixed this oversight.

3. Page 3: “There is a trend toward using an ensemble of methods or methods capable of using higher-order patterns and integrating multiple types of networks (Table 1 ).” Redundant space before “)”.

This has been addressed, thank you.

4. Page 10: “Fig 4. A. Procedure for training and evaluating the XGboost model. Features were generated about each cluster (identified as described in Section )”, which Section? Is the Section number missing here? In page 11: the same issue, “After performing quality control (Section ), there were on average 401 variants per patient (Fig 6.C).”

Missing section numbers have been added, thank you.

5. The author refers g2p to gene to phenotype. Does the author use Phenotype in this paper as a feature (HPO term)? We often use phenotype to refer to disorders and describe HPO terms by features.

No, we do not use HPO terms directly as features. HPO terms can be used as features - that is a valid methodological approach; but that is not the approach used here.

 6. The author's definition of the undiagnosed patient in CHCO and MyGene2 is unclear. The first sentence of the result section states, “We applied BOCC to 721 patients from Children’s Hospital Colorado (CHCO) with suspected genetic drivers of their diseases.” Does that mean these patients are undiagnosed?

Thank you for raising this point of confusion. We do not claim that all 721 CHCO patients are still undiagnosed, as we are not privy to that aspect of the clinical records nor do we have IRB approval to access that information. But we do know

1. They were suspected of having a disease of genetic origin - which is why they underwent WES sequencing

2. There are no known connections between their VUS and the HPO terms assigned by their doctor - a hallmark of an undiagnosed disease case as set forth by the Clinical validity requirement of the CDC’s ACCE model.

We have added clarifying language around this point in a paragraph starting at line 414.

7. Moreover, in MyGene2 section, “We found that presently 111 of these profiles contained no direct connection between any of their genes and any of their HPO terms; we assumed these cases to still be undiagnosed.”, I think it is problematic. It is dangerous to assume the patient is undiagnosed because there is no connection between HPO and the gene. I randomly checked some patients in Table S1 and Table 5. I found some of the patients are already published in the literature, and I believe they are diagnosed. The first question is the patient with ID 3071 (https://mygene2.org/MyGene2/familyprofile/3071/profile). It is the first patient in Table S1. This patient was already published in a paper with PMID (25142838) and annotated with Kabuki and KMT2D. Does the author consider this patient as diagnosed or undiagnosed? For example, patient ID 877 (https://mygene2.org/MyGene2/familyprofile/877/genetic/gene) is already published in https://www.ncbi.nlm.nih.gov/pmc/articles/PMC5082428/. This patient was also annotated with Joubert syndrome and a likely pathogenic mutation in B9D2. Besides, patient ID 1292 has likely pathogenic mutation in TNNT3 and was diagnosed as “Arthrogryposis, Distal, Type 1A”. And the author reported “Arthrogryposis multiplex congenita” as a new relationship to TNNT3. However, we can see from the disease name linked to TNNT3, which is “Arthrogryposis, distal, type 2B2,” and the entry in omim was updated in 2019. I would say this new relationship between this HPO and TNNT3 is not surprised at all because you can see it from the disease name. Therefore, I wonder what the meaning of this prediction is.

The fact that some of these MyGene2 cases have been solved and published is a vote of confidence for our tool; it shows that our predictions were correct! The bigger question is, if these cases are solved, is there no recorded association between them and the rare disease databases on which the edges in HPO are based? That is a question for the database and ontology curators.

 We conceded that our assumption of undiagnosed in MyGene2 cases has limitations and have added a discussion of the points raised by reviewer 5 to the manuscript, see the new paragraph starting at line 414.

The reviewer has struck upon several outstanding issues, limitations but also intentional features of phenotype-based methods and biological ontologies more broadly:

Distinct meaning - “Arthrogryposis, Distal, Type 1A” is a different disease than “Arthrogryposis, distal, type 2B2”. So while it may not be surprising that mutations to the same gene cause them, it would still be a new connection.

Specificity of terms/incompleteness of data - “Arthrogryposis multiplex congenita” is a leaf node in HPO - it is the most specific identifier currently available. Having more general terms like this one is beneficial in that it allows inferences to be made about a class phenotypes more broadly (like what is happening to patient 1292) or to catch cases where more precise terms do not exist and where terms of greater detail need to be added.

We have added some discussion about this topic at line 406, which highlights the need to continue expanding and improving these ontologies.

8. My main concern is that I don’t know how I validate wheter the connection found by the cluster is correct or not. The experiement should start from validate on the existing relationship from diagnosed patients. Then we know that the relationship found in the cluster do have some meanings. Then, we might select some patients with a disease gene found after the model is built. For example, the model trained on the data from 2019, then we can check whether the model can predict the relationship between the HPO and a Gene X, which was found as a disease-causing gene in 2020. As pointed out previously, when the definition of diagnosed and undiagnosed is unclear, it is very difficult to validate whether the cluster is meaningful or not, even though the experiments reported many significant p-values from different methods.

Yes, what you described is exactly how our validation experiments work. We identify clusters on the 2019 data, then label those clusters according to if new edges in the 2020 graph are found with in the 2019 clusters. These clusters and labels are then used to train our machine learning classifiers which are then tested and evaluated on the 2021 data - showing that the procedure works. Unfortunately, we are unaware of a set of patient profiles that would enable such an experiment; but we are open to suggestions.

9. In the end, how the clinicians use this tool is unclear. There are many nodes in the graph, and many clusters were reported after the HPO and genes are given as input. It will be great to show more examples and the steps how the user should use. For example, I tried several HPOs linked to Cornelia de Lange syndrome (NIPBL gene), Long eyelashes (HP:0000527). Synophrys

 (HP:0000664) and Highly arched eyebrow (HP:0002553). However, I only found NIPBL is the third claster. I believe it is the key features for CdLS (NIPBL). Therefore, how do I interpret the first two clusters without NIPBL? The experiment and results showed that this method is working, statistically. However, how to interpret the results and validate the clinical meaning is unclear.

Thank you for taking the time to use our tool! Since there are already known connections between these HPO terms and gene, our tool is of limited utility as it is intended to generate hypotheses about latent gene-to-phenotype connections, regardless we hope you enjoyed the visual interface.

It is odd NIPBL does not show up until cluster 3 for you. When we searched for all those HPO terms and NIPBL, it is present in the first cluster (see screenshot below). As for the interpretation of the first two clusters not having the gene in them, this is likely an artifact of the sorting procedure in the web interface. It sorts by the number of search term matches, then by the predicted p-value, and then breaks ties by sorting the cluster IDs alphabetically.

We have added additional instructions of the web app use at line 284; thank you for the recommendation.

 10. Will the gene in the same pathway or same phenotypic series shown in the same cluster?

Excellent question. Sometimes, but not always. Clustering algorithms are often used for identifying/ inferring genetic pathways and “disease modules” - or groups of closely related diseases. This was part of the motivation for using clustering. We this is mentioned in the introduction, but we have added text highlighting this later on at line 396.

11. In author’s first example, Patient 1930 with VUS in NBEA and SSPO. The author stated in the introduction that with more experiment to the associated phenotype in the clusters could help us to diagnose this patient. I believe the phenotype here is the HPO term. I wonder why we can use the new gene-to-hpo to solve the case with VUS? Does it contribute to the ACMG variant classification?

Nice observations. Yes, the phenotype and the HPO term are the same; we consider these terminologies more or less equivalent the difference being that HPO terms are precise and discrete and may not fulling encompass or describe all phenotypes. A predicted potential link between an HPO term and a VUS does not constitute a solved case; it is a promising hypothesis to be tracked down with further experimentation, literature, and or database research inorder to satisfy ACMG criteria. Our language around this was admittedly imprecise and has been updated, lines 69.

---

## [Decision Letter · Decision Letter 2]

8 Aug 2024

HYPOTHESIS GENERATION FOR RARE AND UNDIAGNOSED DISEASES THROUGH CLUSTERING AND CLASSIFYING TIME-VERSIONED BIOLOGICAL ONTOLOGIES

PONE-D-23-41162R2

Dear Dr. Layer,

We’re pleased to inform you that your manuscript has been judged scientifically suitable for publication and will be formally accepted for publication once it meets all outstanding technical requirements.

Kind regards,

Gary S. Stein

Academic Editor

PLOS ONE

Additional Editor Comments (optional):

Reviewers' comments:

Reviewer's Responses to Questions

**Comments to the Author**

1. If the authors have adequately addressed your comments raised in a previous round of review and you feel that this manuscript is now acceptable for publication, you may indicate that here to bypass the “Comments to the Author” section, enter your conflict of interest statement in the “Confidential to Editor” section, and submit your "Accept" recommendation.

Reviewer #5: All comments have been addressed

2. Is the manuscript technically sound, and do the data support the conclusions?

Reviewer #5: Yes

3. Has the statistical analysis been performed appropriately and rigorously? 

Reviewer #5: Yes

4. Have the authors made all data underlying the findings in their manuscript fully available?

Reviewer #5: Yes

5. Is the manuscript presented in an intelligible fashion and written in standard English?

Reviewer #5: Yes

6. Review Comments to the Author

Reviewer #5: (No Response)

7. PLOS authors have the option to publish the peer review history of their article (what does this mean?). If published, this will include your full peer review and any attached files.

Reviewer #5: **Yes: **Tzung-Chien Hsieh

---

## [Editor Report · Acceptance letter]

23 Aug 2024

PONE-D-23-41162R2 

PLOS ONE

Dear Dr. Layer, 

I'm pleased to inform you that your manuscript has been deemed suitable for publication in PLOS ONE. Congratulations! Your manuscript is now being handed over to our production team.

Kind regards, 

on behalf of

Dr. Gary S. Stein 

Academic Editor

PLOS ONE